# Detecting Contextual Hallucinations in Large Language Models with Frequency-Aware Attention

Siya Qi [1]  Yudong Chen [2]  Runcong Zhao [1]  Qinglin Zhu [1]  Zhanghao Hu [1]  Wei Liu [1]
Yulan He [1 3]  Zheng Yuan [4 3]  Lin Gui [1]

## Abstract

Hallucination detection is critical for ensuring the reliability of large language models (LLMs) in context-based generation. Prior work has explored intrinsic signals available during generation, among which attention offers a direct view of grounding behavior. However, existing approaches typically rely on coarse summaries that fail to capture fine-grained instabilities in attention. Inspired by signal processing, we introduce a frequency-aware perspective on attention by analyzing its variation during generation. We model attention distributions as discrete signals and extract high-frequency components that reflect rapid local changes in attention. Our analysis reveals that hallucinated tokens are associated with high-frequency attention energy, reflecting fragmented and unstable grounding behavior. Based on this insight, we develop a lightweight hallucination detector using high-frequency attention features. Experiments on the RAGTruth and HalluRAG benchmarks show that our approach achieves performance gains over verification-based, internal-representation-based, and attention-based methods across models and tasks.[1]

## 1. Introduction

Large Language Models (LLMs) have achieved strong performance across many natural language processing tasks, yet they can produce *hallucinated outputs* that are fluent

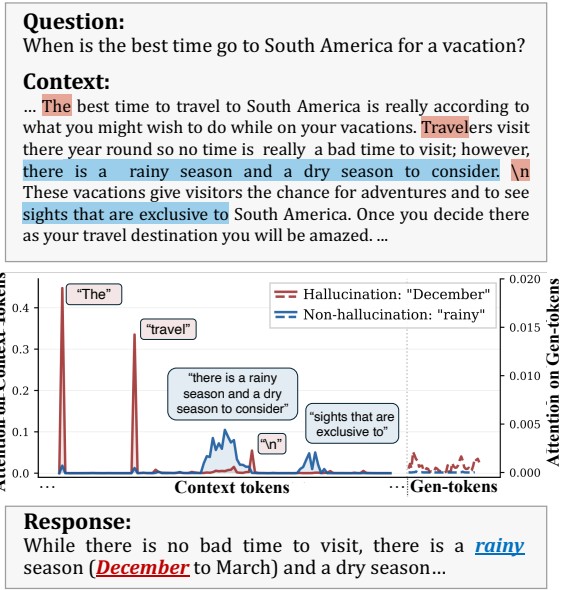

*Figure 1.* Attention weights over context and previously generated tokens for a grounded token (blue, "rainy") and a hallucinated token (red, "December") in a context-based QA example.

[1]Department of Informatics, King's College London, UK [2]Department of Statistics, University of Warwick, UK [3]The Alan Turing Institute, UK [4]School of Computer Science, The University of Sheffield, UK. Correspondence to: Lin Gui <lin.1.gui@kcl.ac.uk>, Siya Qi <siya.qi@kcl.ac.uk>.

*Proceedings of the 43rd International Conference on Machine Learning*, Seoul, South Korea. PMLR 306, 2026. Copyright 2026 by the author(s).

[1]Code and data are available at https://github.com/siyaqi/FrequencyAwareHallucination.

but not supported by the given input or external facts (Ji et al., 2023; van Deemter, 2024). This issue is especially pronounced in the **context-based generation** settings, such as the summarization task, where models are explicitly expected to ground in a provided source context (Hu et al., 2024a). Therefore, effective hallucination detection is essential for building trustworthy language systems and enabling downstream mitigation strategies.

A common approach to hallucination detection verifies model outputs against the source context, for example, using semantic similarity measures or LLM-as-a-judge frameworks (Kryscinski et al., 2020; Laban et al., 2022; Manakul et al., 2023). Such methods compare the output with contextual evidence and are applied post hoc, rather than reflecting the model's generation dynamics. Motivated by this limitation, recent work has explored hallucination detection from intrinsic signals available during generation, including token

probabilities, hidden representations, and attention patterns (Sun et al., 2025; Chen et al., 2024; Chuang et al., 2024a). Among these signals, attention is particularly informative, as it directly reflects how the model allocates content focus during generation (Wiegreffe & Pinter, 2019). Prior studies have shown that hallucinated generations are often associated with unstable or diffuse attention behavior (Liu et al., 2023; Gong et al., 2024; Sriramanan et al., 2024). However, *how to quantify attention stability or uncertainty remains an open problem.*

Most existing attention-based methods summarize attention using coarse statistics, such as attention mass, entropy, or transition patterns (Huang et al., 2025; Sun et al., 2025; Chuang et al., 2024a). While effective at capturing overall concentration, such scaling-based metrics often discard fine-grained sequential variation. For example, in Figure 1, we prompt LLaMA-2-7B-Chat with a context-based question and require the model to answer strictly using the provided context. Although most generated tokens align well with the source, the model produces a hallucinated token containing month names (e.g., "December") that does not appear in the context. Compared to a grounded token, the attention distribution associated with this hallucinated token exhibits noticeably stronger local fluctuations across context positions, with sharper peaks and more abrupt changes.

Therefore, we argue that directly mapping an attention sequence to a single scalar cannot fully capture the structure of attention patterns. Inspired by signal processing, we treat attention weights over context tokens as a discrete temporal signal indexed by token position, where stable grounding corresponds to smooth, slowly varying signals, while instability manifests as rapid local oscillations. To explicitly quantify such variation, we perform frequency-aware decomposition of the attention signal: low-frequency components capture global trends of attention allocation across the context, whereas high-frequency components isolate sharp spikes and abrupt local changes. We hypothesize that hallucinated tokens are associated with energy in these high-frequency attention components.

Motivated by this perspective, we study contextual hallucination detection via frequency-aware analysis of attention signals. Our contributions are threefold: **(1)** We formulate attention as discrete signals and introduce a unified frequency-based framework for analyzing attention variation for hallucination detection. **(2)** We instantiate this framework using simple yet efficient high-frequency extraction operators, enabling the quantification of attention instability at both token and span levels. **(3)** Through extensive experiments across multiple models and tasks, we show that frequency-based attention features can improve hallucination detection over existing verification-based, internal representation-based, and attention-based baselines.

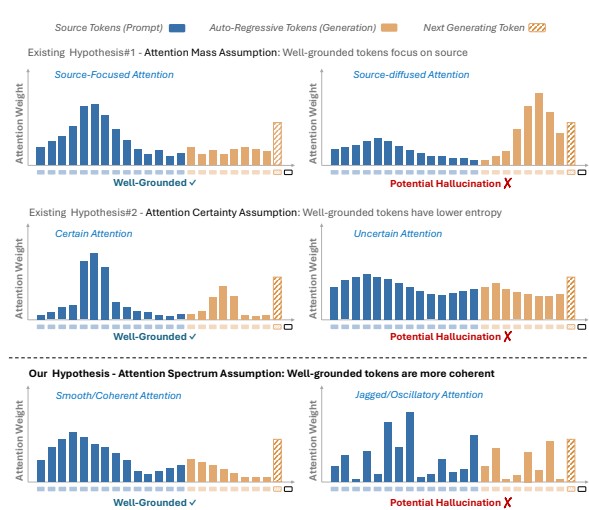

*Figure 2.* Three hypotheses for identifying hallucination tokens from incoming attention patterns. We illustrate three representative assumptions for distinguishing well-grounded tokens (✓) from potential hallucinations (✗) based on the incoming attention to the next generated token.

## 2. Background

In this section, we review attention weight as an intrinsic signal for hallucination and introduce a frequency-based perspective that characterizes attention instability beyond the coarse allocation statistics.

### 2.1. Attention-based Hallucination Detection

For each generated token, the attention distribution reflects how the model aggregates information from the source context and previously generated tokens, and is therefore widely used as an intrinsic signal of grounding and information usage (Campbell et al., 2023; Li et al., 2023; Snyder et al., 2024; Huang et al., 2025). Prior work exploits attention for hallucination detection by making different assumptions about grounded generation behavior.

One line of work focuses on attention transitions, using backward attention from generated tokens to the source context as an indicator of grounding (Chuang et al., 2024a; Ogasa & Arase, 2025). Another line analyzes the distributional properties of attention, either by identifying context tokens emphasized by specific attention heads (Sun et al., 2025) or by summarizing attention uncertainty using entropy-based measures (Vazhentsev et al., 2025). These approaches are motivated by the intuition that grounded generation exhibits focused and consistent evidence attribution, whereas hallucination is associated with diffuse or ambiguous attention.

As illustrated in Figure 2, existing attention-based methods

primarily capture *where* attention is allocated or *how* dispersed it is, but do not explicitly model the internal structure of attention patterns. In contrast, our hypothesis emphasizes the *coherence* of attention distributions, motivated by the observation that hallucinated generations often exhibit fragmented or rapidly oscillating attention behavior that is not captured by static allocation or entropy-based statistics.

## 2.2. A Frequency-based View of Attention

To capture attention variation beyond static summaries, we model attention weights over tokens as discrete signals. In signal processing, smooth and coherent patterns are naturally represented by low-frequency components, whereas abrupt changes and local irregularities manifest as high-frequency components. As such, frequency analysis offers a natural characterization of attention stability and instability.

Rather than committing to a single formulation, we consider several standard operators that instantiate this frequency-based view with complementary inductive biases. The Discrete Fourier Transform (DFT) (Cooley et al., 1969) provides a global decomposition of attention signals into frequency components. The Discrete Wavelet Transform (DWT) (Mallat, 1989) uses localized basis functions, allowing abrupt attention changes to be captured while preserving positional information. As a simpler local alternative, the discrete Laplacian operator (Oppenheim et al., 1997) directly highlights second-order differences directly in the token domain, acting as an implicit high-pass filter.

Despite their different forms, these operators share a common goal: isolating high-frequency variation that reflects fragmented or rapidly oscillating attention behavior. Together, they provide a unified frequency-based framework for analyzing attention instability during generation.

# 3. Frequency-Aware Attention Modeling

Viewing attention from a frequency-aware perspective offers a principled way to analyze variation patterns that are difficult to characterize directly in the token domain. In this framework (shown in Figure 3), attention weights over tokens are modeled as discrete signals, enabling the use of frequency-aware operators to quantify how attention evolves and fluctuates during generation, beyond what is captured by aggregate statistics. We first demonstrate our motivation and intuition through a simplified setting here.

## 3.1. Motivation and Intuition

While a rigorous mechanistic understanding of attention behavior under hallucinated generation remains an open problem, we can gain preliminary insights from a simplified toy setting. Specifically, consider a scenario where tokens come from several latent semantic sources. As the number of such sources increases, neighboring tokens are more likely to differ in topic, causing the compatibility between the current token and its context to change abruptly across adjacent positions. Such abrupt local changes translate into adjacent differences in attention weights, yielding jagged attention patterns along the sequence (see Appendix A for a complete proof). Although such toy analysis relies on simplifying assumptions, it formalizes a general link between semantic heterogeneity and attention instability.

Real LLM attention is, however, substantially far more complex. It is shaped by architectural depth, multi-head specialization, and long-range contextual interactions, all of which can give rise to higher-order and multi-scale instabilities that cannot be captured by adjacent difference measures alone. From a signal-processing perspective, adjacent differences correspond only to a primitive form of high-pass filtering. Frequency-aware operators (e.g., DFT, DWT, Laplacian), by contrast, are able to systematically extract high-frequency components across multiple resolutions (Donoho & Johnstone, 1998; Stein & Shakarchi, 2003). This motivates our frequency-aware framework for quantifying complex attention instabilities associated with hallucination detection.

## 3.2. Problem Setup

We consider a context-based generation setting, where a language model generates a response $\mathbf{gen} = (gen_1, \ldots, gen_T)$ conditioned on a retrieved context $\mathbf{ctx} = (ctx_1, \ldots, ctx_N)$. At each generation step $i$, the model produces token $t_i$ based on $(\mathbf{ctx}, \mathbf{gen}_{<i})$ and exposes attention weights across layers and heads. Our primary task is token-level hallucination detection: for each generated token $t_i$, predict whether $t_i$ is supported by the provided context.

Rather than operating on attention weights directly, our detector uses frequency-aware features derived from attention. For each step $i$ when generating $t_i$, we compute a feature vector $\mathbf{v}$ (defined in §3.4) and predict

$$\hat{r}_i = f(\mathbf{v}), \tag{1}$$

where $f$ is a lightweight linear classifier, and $\hat{r}_i$ indicates a prediction result for hallucination or non-hallucination.

## 3.3. Attention as Discrete Signals

At generation step $i$, we extract attention weight distributions from all transformer layers and heads. We distinguish two types of attention signals: *context-directed attention*, attending from the current token $t_i$ to the input context tokens, and *generated-token attention*, attending to previously generated tokens before this generation step.

For each layer $l \in \{1, \ldots, L\}$ and head $h \in \{1, \ldots, H\}$, we define the context-directed attention vector and the

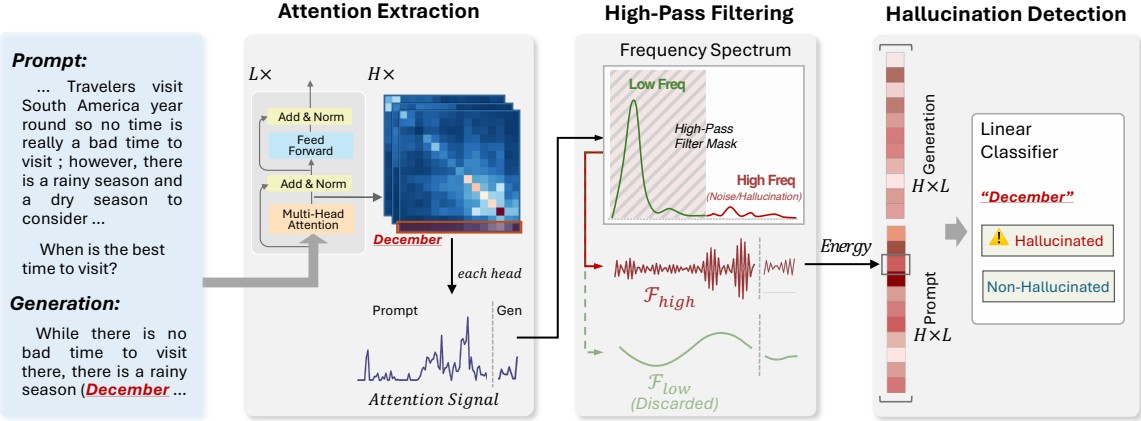

*Figure 3.* Overview of frequency-aware attention modeling for hallucination detection. Attention weights are extracted from each layer and head ($L$ layers and $H$ heads in total), treated as token-level signals, and decomposed using high-pass filtering to isolate high-frequency variations ($\mathcal{F}_{\text{high}}$), whose energy is aggregated for hallucination detection.

generated-token attention vector:

$$
\begin{aligned}
a_{l,h}^{(ctx)} &= [a_{l,h,1}^{(ctx)}, a_{l,h,2}^{(ctx)}, \ldots, a_{l,h,N}^{(ctx)}] \in \mathbb{R}^N, \\
a_{l,h}^{(gen)} &= [a_{l,h,1}^{(gen)}, a_{l,h,2}^{(gen)}, \ldots, a_{l,h,i-1}^{(gen)}] \in \mathbb{R}^{i-1},
\end{aligned}
\tag{2}
$$

which respectively capture attention over $N$ context tokens and the previously generated tokens before $t_i$.

Both $a_{l,h}^{(ctx)}$ and $a_{l,h}^{(gen)}$ are treated as one-dimensional discrete signals indexed by token position. Taking the Discrete Fourier Transform (DFT) as an example, and let $\mathcal{F}$ denote DFT. Mapping the attention weight vector to the frequency domain yields

$$
\hat{a}_{l,h}^{(\tau)} = \mathcal{F}\big(a_{l,h}^{(\tau)}\big),
\tag{3}
$$

where $\tau \in \{ctx, gen\}$; details for the DWT and Laplacian operators are provided in Appendix B.

### 3.4. Energy-Based High-Frequency Instability

Building on the frequency-based view, we isolate and quantify high-frequency components of attention weight signals. The formulation applies uniformly to both context-directed and previously generated-token attention. To simplify notation, we use $\mathbf{x} \in \mathbb{R}^n$ to denote the attention signal obtained for $t_i$, corresponding to either $a_{l,h}^{(ctx)}$ or $a_{l,h}^{(gen)}$ for layer $l$ and head $h$, with signal length $n$.

Given an attention signal $\mathbf{x}$, we extract high-frequency component by high-pass operator (e.g., DFT, DWT, Laplacian)

$$
\mathbf{z}^{\text{hf}} = \mathcal{F}_{\text{high}}(\mathbf{x}),
\tag{4}
$$

where $\mathcal{F}_{\text{high}}$ denotes a high-pass operator that suppresses smooth, low-frequency components while preserving high-frequency components representing rapid local variation.

For transform-based operators such as DFT and DWT, $\mathcal{F}_{\text{high}}$ is implemented by frequency-domain masking followed by an inverse transform. Let $\hat{\mathbf{x}} = \mathcal{F}(\mathbf{x})$ denote the DFT coefficients. We retain only high-frequency components by applying a frequency mask by following definition:

$$
\hat{\mathbf{x}}_k^{\text{hf}} = \begin{cases} \hat{\mathbf{x}}_k, & k \in \mathcal{K}_{\text{high}}, \\ 0, & \text{otherwise}, \end{cases}
\tag{5}
$$

where $\mathcal{K}_{\text{high}}$ indexes the retained high-frequency band. The corresponding high-frequency component $\mathbf{z}^{\text{hf}}$ in the temporal domain is obtained via inverse transform. Wavelet- and Laplacian-based operators provide alternative realizations of $\mathcal{F}_{\text{high}}$ that emphasize localized high-frequency variation, while serving the same purpose of isolating rapid attention fluctuations (detailed in Appendix B).

We summarize the magnitude of high-frequency variation by using the $\ell_2$ norm on the high-frequency component

$$
\rho = \|\mathbf{z}^{\text{hf}}\|_2,
\tag{6}
$$

which measures the energy of high-frequency components in the attention signal in temporal domain.

The use of the $\ell_2$ norm is theoretically motivated by **Parseval's theorem**, which establishes the equivalence between signal energy measured in the temporal domain and that measured in the frequency domain. For DFT-based filtering,

$$
\|\mathbf{z}^{\text{hf}}\|_2^2 = \sum_{j=1}^{n} |\mathbf{z}_j^{\text{hf}}|^2 = \frac{1}{n} \sum_{k=0}^{n-1} |\hat{\mathbf{x}}_k^{\text{hf}}|^2.
\tag{7}
$$

Applying the above procedure independently to each attention head and each attention type yields a score $\rho_{l,h}^{(\tau)}$ for

*Table 1.* Comparison of different methods on RAGTruth and HalluRAG. Wavelet-high and Fourier-high denote leveraging the high-frequency components obtained via DWT and DFT, respectively. **Bold** and underlined values indicate the best and second-best performance within each model group.

| Model | Method | RT-QA | | RT-D2T | | RT-Summ | | HalluRAG | | Overall Avg. | |
|---|---|---|---|---|---|---|---|---|---|---|---|
| | | F1 | AUROC | F1 | AUROC | F1 | AUROC | F1 | AUROC | Avg-F1 | Avg-A |
| LLaMA-7B | SelfCheckGPT | 0.6289 | 0.6942 | 0.6552 | 0.8026 | **0.6349** | 0.6674 | 0.5388 | 0.5963 | 0.6145 | 0.6901 |
| | RefChecker | 0.5914 | 0.5865 | 0.5713 | 0.6349 | 0.6059 | 0.6080 | 0.5819 | 0.5722 | 0.5876 | 0.6004 |
| | EigenScore | 0.4979 | 0.5253 | 0.5256 | 0.5297 | 0.5065 | 0.4989 | 0.5364 | 0.6127 | 0.5166 | 0.5416 |
| | Redeep | 0.5972 | 0.6364 | 0.4791 | 0.3960 | 0.5897 | 0.5760 | 0.5912 | 0.6490 | 0.5643 | 0.5643 |
| | Lookback-lens | 0.6930 | 0.8482 | 0.6175 | 0.8442 | 0.5328 | 0.7156 | 0.6266 | 0.7405 | 0.6175 | 0.7871 |
| | Attn-variance | 0.4807 | 0.6147 | 0.4839 | 0.5890 | 0.4886 | 0.6492 | 0.4489 | 0.5571 | 0.4755 | 0.6025 |
| | Attn-entropy | 0.6832 | 0.8481 | 0.6011 | 0.8368 | 0.5031 | 0.6722 | 0.6020 | 0.6937 | 0.5973 | 0.7627 |
| | Laplacian | 0.7107 | 0.8449 | 0.6878 | 0.8519 | 0.5779 | 0.7040 | 0.6370 | 0.7429 | 0.6534 | 0.7859 |
| | Wavelet-high | 0.7194 | 0.8526 | **0.6898** | 0.8569 | 0.5929 | 0.7165 | 0.6384 | 0.7550 | 0.6601 | 0.7953 |
| | Fourier-high | **0.7277** | **0.8584** | 0.6870 | **0.8595** | 0.5875 | **0.7426** | **0.6438** | **0.7603** | **0.6615** | **0.8052** |
| LLaMA-13B | SelfCheckGPT | 0.6029 | 0.6425 | 0.6346 | 0.7989 | 0.5554 | 0.5909 | 0.6337 | 0.7216 | 0.6066 | 0.6885 |
| | RefChecker | 0.5928 | 0.5893 | 0.5784 | 0.6381 | 0.5924 | 0.6346 | 0.6098 | 0.5963 | 0.5934 | 0.6146 |
| | EigenScore | 0.4788 | 0.4734 | 0.6063 | 0.6582 | 0.5121 | 0.5097 | 0.6316 | 0.6947 | 0.5572 | 0.5840 |
| | Redeep | 0.5740 | 0.6457 | 0.5273 | 0.6449 | 0.5010 | 0.5253 | 0.5769 | 0.6294 | 0.5448 | 0.6113 |
| | Lookback-lens | 0.6947 | 0.8727 | 0.7137 | 0.8766 | 0.5679 | 0.6702 | 0.6594 | **0.7929** | 0.6589 | 0.8031 |
| | Attn-variance | 0.5636 | 0.7497 | 0.5992 | 0.7177 | 0.4947 | 0.6629 | 0.4529 | 0.6298 | 0.5276 | 0.6900 |
| | Attn-entropy | 0.6909 | 0.8650 | 0.7034 | 0.8717 | 0.5527 | 0.6865 | 0.6284 | 0.7270 | 0.6438 | 0.7875 |
| | Laplacian | 0.6834 | 0.8552 | 0.7194 | 0.8796 | 0.5548 | 0.6700 | 0.6659 | 0.7624 | 0.6559 | 0.7918 |
| | Wavelet-high | 0.7029 | 0.8741 | **0.7383** | **0.8932** | 0.5651 | 0.7042 | **0.6684** | 0.7809 | 0.6687 | 0.8131 |
| | Fourier-high | **0.7068** | **0.8792** | 0.7278 | 0.8825 | **0.5929** | **0.7362** | 0.6616 | 0.7899 | **0.6723** | **0.8219** |
| Mistral-7B | SelfCheckGPT | 0.6551 | 0.7084 | 0.6958 | 0.8353 | **0.6966** | 0.7166 | 0.5515 | 0.6473 | 0.6498 | 0.7269 |
| | RefChecker | 0.6144 | 0.6121 | 0.5907 | 0.6287 | 0.6664 | 0.6596 | 0.6176 | 0.6031 | 0.6223 | 0.6259 |
| | EigenScore | 0.4904 | 0.4611 | 0.5449 | 0.6507 | 0.4582 | 0.4973 | 0.6581 | 0.7924 | 0.5379 | 0.6004 |
| | Redeep | 0.6270 | 0.7357 | 0.6013 | 0.6648 | 0.5379 | 0.5680 | 0.4987 | 0.5646 | 0.5662 | 0.6333 |
| | Lookback-lens | 0.7832 | **0.9148** | 0.7151 | 0.8845 | 0.6759 | 0.7954 | 0.7115 | 0.7966 | 0.7214 | 0.8478 |
| | Attn-variance | 0.5636 | 0.7497 | 0.6218 | 0.7021 | 0.4840 | 0.5836 | 0.5367 | 0.6953 | 0.5515 | 0.6827 |
| | Attn-entropy | 0.7810 | 0.9083 | 0.7049 | 0.8734 | 0.6551 | 0.7867 | 0.6803 | 0.7504 | 0.7053 | 0.8297 |
| | Laplacian | 0.7807 | 0.9049 | 0.7255 | 0.8849 | 0.6723 | 0.7978 | 0.7001 | 0.8098 | 0.7197 | 0.8494 |
| | Wavelet-high | 0.7876 | 0.9117 | 0.7136 | 0.8829 | 0.6849 | **0.8075** | **0.7274** | **0.8360** | **0.7284** | **0.8595** |
| | Fourier-high | **0.7885** | 0.9099 | **0.7267** | **0.8863** | 0.6761 | 0.8037 | 0.7152 | 0.8128 | 0.7266 | 0.8532 |

every layer $l$, head $h$, and attention type $\tau \in \{ctx, gen\}$.

$$\mathbf{v}^{(\tau)} = [\rho_{1,1}^{(\tau)}, \rho_{1,2}^{(\tau)}, \dots, \rho_{L,H}^{(\tau)}]^\top, \quad \tau \in \{ctx, gen\},$$
$$\mathbf{v} = [\mathbf{v}^{(ctx)}; \mathbf{v}^{(gen)}]. \tag{8}$$

We aggregate these scores across all layers and heads to form feature vectors $\mathbf{v}^{(ctx)}$ and $\mathbf{v}^{(gen)}$ for context-directed and generated-token attention, respectively. We then concatenate these vectors as $\mathbf{v}$ and feed them into the classifier to obtain the final prediction $\hat{r}_i$ for token-level hallucination detection. As a robustness check under coarser supervision, we also report span-level results by applying a fixed sliding window, averaging the feature vectors within each window, and feeding the averaged vectors into the classifier.

## 4. Experiment Setting

### 4.1. Baselines

In this study, we evaluate our approach against a diverse set of representative baselines for comparison, spanning verification-based, internal representation-based, and attention-based paradigms, as detailed below.

**Verification-based Methods.** **SelfCheckGPT** (Manakul et al., 2023) detects hallucinations by measuring stochastic consistency across multiple responses sampled from a language model. **RefChecker** (Hu et al., 2024b) extracts claims from model outputs and verifies them against context using a dedicated checker. Both of them operate in a prompt-based manner and rely on the generative behavior of the underlying LLM.

**Internal Representation-Based Methods.** **EigenScore** (Chen et al., 2024) leverages output probability distributions to construct a semantic consistency graph and quantifies uncertainty via spectral analysis. **ReDeEP** integrates internal signals from hidden states and attention mass to detect hallucinations (Sun et al., 2025). Both methods represent strong internal-signal-based baselines.

**Attention-based Methods.** We include **Lookback-Lens** (Chuang et al., 2024a) as a strong attention-based baseline,

which characterizes hallucination by quantifying the allocation of attention between retrieved context tokens and previously generated tokens. In addition, we implement two mechanistic attention-based baselines based on attention **variance** and attention **entropy**. For these baselines, statistics are computed separately over context tokens and generated tokens, and concatenated across all attention heads to form a unified feature representation for classification.

We note that SelfCheckGPT, RefChecker, EigenScore, and ReDeEP are unsupervised methods that do not require hallucination annotations for training, whereas our method uses labeled hallucination data to calibrate the final classifier. Nevertheless, these methods remain strong and widely used baselines that represent complementary hallucination detection paradigms. To further improve comparability, we additionally evaluate unsupervised setting of our method in Appendix F.3.

### 4.2. Implementation Settings

To evaluate the effectiveness and robustness of our method, we conduct experiments across three widely used open-source LLMs, including LLaMA-7B-Chat, LLaMA-13B-Chat, and Mistral-7B-Instruct, covering diverse model sizes and architectures. Experiments are conducted on two context-based hallucination benchmarks: RAGTruth (Niu et al., 2024), covering question answering (QA), data-to-text (D2T), and summarization (Summ), and HalluRAG (Ridder & Schilling, 2024), which focuses on the QA task. Both datasets provide token-level hallucination annotations. Frequency-aware attention features are aggregated using a single-layer logistic regression classifier, and detection is performed at either the token or span level. We report F1 score and AUROC on the test sets, using AUROC as the primary metric. Additional details are provided in Appendix C.3.

## 5. Results and Analysis

### 5.1. Overall Performance

Across all evaluated models and datasets, frequency-aware analysis of attention consistently improves hallucination detection over verification-based, internal representation-based, and attention-based baselines. Explicitly isolating high-frequency components of attention yields stronger performance than aggregate statistics such as variance or entropy, highlighting the importance of modeling fine-grained attention variation. By focusing on *how* attention varies within the sequence rather than *where* attention mass is allocated, frequency-aware features provide complementary discriminative signals beyond attention mass alone.

These improvements hold across diverse task formats and models. For example, on summarization task, Fourier-

high improves AUROC by 6.6% over Lookback-Lens on LLaMA-13B, and by 2.7% on LLaMA-7B. This consistency across datasets and generation settings suggests that high-frequency attention variation captures a general property of ungrounded generation, rather than task- or structure-specific artifacts. We further evaluate cross-domain generalization by training the detector on one task and testing it on another (see Table A12). Compared to Lookback-Lens, our method exhibits substantially more robust transfer performance, indicating that frequency-aware attention features generalize better across task boundaries.

Among all three operators, Fourier-based features achieve the strongest overall performance, followed by wavelets and the Laplacian. This ordering aligns with both their inductive biases and the structural properties of attention signals: attention distributions are typically sparse and highly uneven across positions (Nawrot et al., 2025), often dominated by a small subset of tokens, which favors operators that aggregate high-frequency variation globally. Correspondingly, Fourier-based filtering is particularly effective at capturing global high-frequency energy, while wavelets and the Laplacian emphasize progressively more localized variation.

*Table 2.* Performance comparison on RagTruth-Avg and HalluRAG for sliding-window=8. Within each model, the best result is highlighted in **bold**, and the second-best is underlined.

| Model | Method | RagTruth-Avg | | HalluRAG | |
|---|---|---|---|---|---|
| | | F1 | AUROC | F1 | AUROC |
| LLaMA-7B | Lookback-lens | 0.6773 | 0.7884 | 0.6623 | 0.7856 |
| | Laplacian | 0.6747 | 0.7877 | 0.6775 | 0.7754 |
| | Wavelet-High | 0.6911 | 0.8117 | 0.6812 | 0.7928 |
| | Fourier-high | **0.7003** | **0.8412** | **0.6866** | **0.8100** |
| LLaMA-13B | Lookback-lens | 0.6781 | 0.8183 | 0.7096 | 0.8505 |
| | Laplacian | 0.6819 | 0.8039 | 0.7217 | 0.8264 |
| | Wavelet-High | 0.6953 | 0.8448 | 0.7417 | 0.8438 |
| | Fourier-high | **0.7063** | **0.8585** | 0.7217 | **0.8515** |
| Mistral-7B | Lookback-lens | 0.7554 | 0.8764 | 0.7752 | 0.8403 |
| | Laplacian | 0.7322 | 0.8495 | **0.8056** | **0.8920** |
| | Wavelet-High | 0.7597 | 0.8742 | 0.7682 | 0.8802 |
| | Fourier-high | **0.7663** | **0.8812** | 0.7883 | 0.8872 |

### 5.2. Span-level Hallucination Detection

In practical settings, hallucinations often occur as contiguous spans rather than isolated tokens. To evaluate whether our method generalizes beyond token-level detection, we conduct span-level hallucination detection under a sliding-window setting, where consecutive tokens are grouped into fixed-size chunks and classified at the chunk level. Following prior work such as Lookback-Lens, we use a chunk size of 8 and aggregate attention features within each chunk to form a single representation for prediction. Results across the two benchmarks are reported in Table 2, with full per-task results provided in Table A2 for RAGTruth.

Span-level evaluation is more challenging due to coarser

supervision and the need to aggregate signals across multiple tokens. Despite this increased difficulty, Fourier- and Wavelet-based variants achieve the strongest performance on all three tasks. Fourier-based features yield consistent gains over Lookback-Lens, improving AUROC by 5.3% on LLaMA-7B (RAGTruth-Avg), with a particularly improvement of 10.1% on the summarization task (see Table A2). Overall, these results indicate that frequency-based features remain robust when individual token-level signals are aggregated into spans, and that our approach generalizes naturally from token- to span-level hallucination detection across multiple tasks without requiring architectural modification.

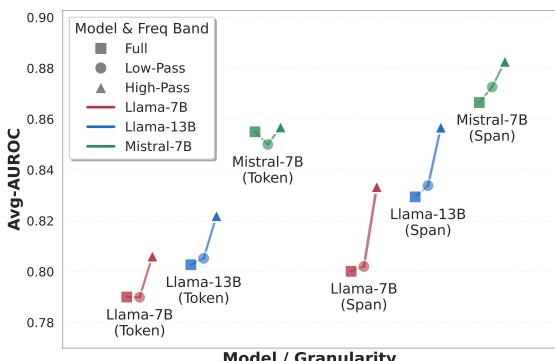

*Figure 4.* Comparing full-, low-, and high-pass Fourier attention features. Average AUROC across models under token- and span-level evaluation settings.

## 5.3. Ablation Study on High/Low-pass Components

We further analyze the contribution of different frequency bands by comparing low-pass, high-pass, and full-spectrum attention features, as shown in Figure 4. The reported Average AUROC is computed across all evaluation datasets (RAGTruth-QA, RAGTruth-D2T, RAGTruth-Summ, and HalluRAG) under both token- and span-level settings. Across all models and evaluation settings, high-pass components achieve the strongest performance, while low-pass and full-spectrum features perform comparably. This pattern suggests that, without explicitly isolating high-frequency variation, attention signals are largely dominated by low-frequency components that offer limited discriminative power for hallucination detection.

Low-frequency components mainly reflect smooth, global alignment patterns common to both grounded and hallucinated outputs, whereas high-pass components emphasize rapid local irregularities in attention that are more closely associated with unstable generation behavior. Additional analyses on Fourier frequency cutoffs and Wavelet detail levels are provided in Figure A1, Table A4, and Table A5.

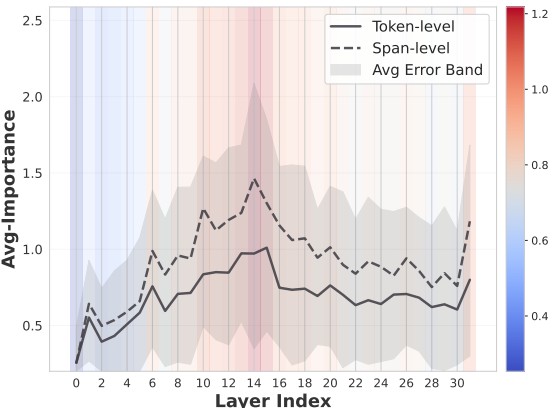

*Figure 5.* Layer-wise importance of high-frequency Fourier-high attention features for LLaMA-7B. The shaded region indicates the standard deviation of head-level importance within each layer. The color bar provides a visualization of the averaged importance values shown on the left y-axis to facilitate layer-wise comparison.

## 5.4. Understanding Frequency-Aware Features

To better understand how frequency-aware attention features contribute to hallucination detection, we analyze the learned linear classifier from three perspectives: layer-level importance, head-level sparsity, and the relative contribution of context versus generated attention.

### 5.4.1. LAYER-WISE IMPORTANCE

As shown in Figure 5, we report the average absolute classifier coefficients assigned to attention heads in each layer. The importance of high-frequency attention signals is clearly non-uniform across model depth. For Fourier-based features, importance peaks in the middle layers, with a pronounced maximum around layer 14. These layers are commonly associated with higher-level semantic processing and factual reasoning, suggesting that hallucination-related attention instability emerges most prominently once the model gradually transitions from surface-level decoding to semantic composition (Chuang et al., 2024b).

Across almost all layers, span-level (dashed line) assigns higher importance weights than token-level detection (solid line). This indicates that aggregating attention signals over longer spans amplifies structured high-frequency variation, rather than weakening it, further supporting the robustness of frequency-aware features under coarser supervision.

### 5.4.2. HEAD-WISE SPARSITY

Beyond layer-level trends, we examine whether detection signals are broadly distributed across heads or concentrated in a small subset. To this end, we perform detection using only the Top-$k$ attention heads ranked by their average

absolute classifier coefficients.

*Table 3.* AUROC Results for LLaMA-7B: Original vs. Top-$k$ head only performance.

| RAGTruth-Avg | | | | |
|---|---|---|---|---|
| **Method** | **Original** | **Top-$k$ heads** | | |
| | | $k = 100$ | 50 | 10 |
| Laplacian | 0.8003 | 0.7824 | 0.7666 | 0.7131 |
| Wavelet-high | 0.8087 | 0.7900 | 0.7708 | 0.7193 |
| Fourier-high | 0.8205 | 0.7915 | 0.7684 | 0.6995 |
| **HalluRAG** | | | | |
| **Method** | **Original** | **Top-$k$ heads** | | |
| | | $k = 100$ | 50 | 10 |
| Laplacian | 0.7429 | 0.7222 | 0.6986 | 0.6479 |
| Wavelet-high | 0.7550 | 0.7229 | 0.7016 | 0.6562 |
| Fourier-high | 0.7629 | 0.7229 | 0.6814 | 0.6373 |

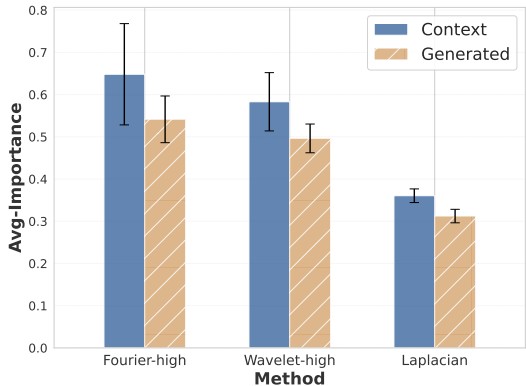

*Figure 6.* Average absolute classifier importance assigned to context-attention and generated-attention features over all examined models and datasets.

As shown in Table 3, a remarkably small fraction of heads accounts for most of the detection performance. Using only the top 100 heads (less than 10% of all heads in LLaMA-7B) recovers 95% of the original AUROC across operators and datasets. This pronounced sparsity indicates that hallucination-related high-frequency attention variation is not a diffuse property shared across all heads. Instead, a small subset of heads consistently exhibits strong sensitivity to attention instability, suggesting that these heads act as implicit internal indicators of ungrounded generation.

### 5.4.3. CONTEXT VS. GENERATED ATTENTION CONTRIBUTION

A critical design choice in our method is incorporating high-frequency attention signals from both source context tokens and previously generated tokens during generation. We analyze the learned classifier to assess the relative importance of these two sources in practice, where feature importance is measured by the average absolute coefficient magnitude associated with each feature group.

As shown in Figure 6, frequency-aware signals derived from context-token attention are consistently more informative for hallucination detection than those from generated-token attention across all spectral operators. Among them, the Fourier-based method exhibits the largest context–generated importance gap, which aligns with its superior overall performance across tasks and models, suggesting that capturing frequency-aware patterns in context attention contributes to more robust hallucination detection.

This asymmetry aligns closely with the core motivation of Lookback-lens (Chuang et al., 2024a), which emphasizes the role of backward attention to contextual evidence in grounded generation. Our results extend this view by showing that not only the presence of backward attention, but

also its stability and variation, play a critical role in practice. This interpretation is further supported by more ablation results in Table A7, where removing only context-based features leads to substantially larger performance degradation than removing only generated-token features.

## 6. Broader Related Work

In this section, we briefly situate our work within the broader literature on hallucination detection in LLMs.

A substantial line of work detects hallucinations through **external knowledge verification**, validating model outputs against curated corpora, knowledge bases, or web search results (Min et al., 2023; Feng et al., 2023; Chern et al., 2023; Qi et al., 2025a). While effective when reliable evidence is available, these approaches are constrained by knowledge coverage, retrieval quality, and inference-time latency, limiting their applicability in on-the-fly detection settings.

In contrast, **intrinsic detection** methods rely solely on model-internal signals and are lightweight enough to operate during decoding. Prior work has explored semantic consistency between input and output, uncertainty and self-consistency across generations, and generation-time statistics such as token probabilities or hidden-state geometry (Qi et al., 2025b). More recent studies probe attention mechanisms as intrinsic indicators of hallucination (Sun et al., 2025; Chuang et al., 2024a; Campbell et al., 2023), which mainly characterize where attention is allocated or how concentrated it is, but largely overlook fine-grained local variation within the token sequence.

Using **frequency-aware** tools offers an additional perspective for examining internal model dynamics (Kiruluta, 2025; Li et al., 2025). By modeling internal signals as discrete sequences, frequency-aware representations capture struc-

tural patterns that are difficult to characterize using token-level statistics alone. Attention distributions are also typically sparse and highly non-uniform across token positions (Nawrot et al., 2025; Gu et al., 2025b), providing a natural basis for frequency-based decomposition, where abrupt allocation correspond to high-frequency components and global distribution patterns align with low-frequency trends.

Building on this perspective, our work introduces a frequency-aware characterization of attention variation as an intrinsic signal for hallucination detection.

## 7. Conclusion

This work presents a frequency-aware perspective for analyzing hallucination in LLMs. We show that hallucinated generation is associated with high-frequency variation in attention patterns, revealing a structural property of model behavior that is not captured by attention allocation, transition, or aggregate uncertainty measures. This provides a principled way to distinguish grounded from ungrounded generation based on how attention varies across tokens.

Building on this insight, we introduce a frequency-aware attention modeling framework that extracts high-frequency attention signals using spectral operators. Across multiple models and tasks, our approach improves hallucination detection at both token and span levels, indicating its potential as a generalizable and task-agnostic intrinsic signal for analysis. Overall, our findings suggest that hallucinations are reflected not only in where attention is placed, but also in how it varies across tokens over time, highlighting frequency-aware analysis as a promising direction for diagnosing and mitigating unreliable LLM generation.

## Limitations

Despite the improved empirical performance and interpretability of our approach, several limitations remain in both the theoretical assumptions and practical deployment settings.

Our theoretical analysis is developed under a simplified setting with independent token embeddings to illustrate the connection between semantic heterogeneity and high-frequency attention variation. While this provides intuitive justification, it does not fully capture the autoregressive of LLM generation, and extending the analysis to more realistic dependency structures remains future work.

Our span-level detector aggregates token-level signals using sliding-window averaging. Although this follows standard practice in prior hallucination detection work and empirically improves span-level robustness, the aggregation may smooth the high-frequency feature that motivate our method to some extent, representing a practical trade-off between local sensitivity and robust span-level prediction.

From a practical perspective, our approach requires access to internal attention weights and therefore cannot be directly applied to API-only models. In addition, frequency transforms of attention introduce extra computation compared to scalar-based attention summaries. Although our method remains substantially cheaper than multi-generation verification approaches, improving efficiency and applicability to closed-source systems remains future work.

Finally, our detector still relies on hallucination annotations to train the final classifier. Although our experiments demonstrate relatively strong cross-domain transfer and partially unsupervised variants, performance still degrades under substantial distribution shifts across tasks and benchmarks. Reducing dependence on task-specific supervision remains an important challenge for practical deployment.

## Acknowledgements

SQ, QZ, ZH are funded by a PhD scholarship provided by K-CSC. YH was supported by the UK Engineering and Physical Sciences Research Council (EPSRC) through a Turing AI Fellowship (grant no. EP/V020579/1, EP/V020579/2). The authors acknowledge the use of King's Computational Research, Engineering and Technology Environment (CREATE) at King's College London.

## Impact Statement

This paper presents a frequency-aware analysis of internal attention in LLMs for detecting hallucinated generation. The goal of this work is to advance the understanding and evaluation of LLMs by providing intrinsic signals that can help identify unreliable outputs.

There are potential positive societal impacts of this work, including improved reliability and transparency of language model deployments in applications where factual correctness is important. By relying solely on model-internal signals, the proposed approach avoids dependence on external knowledge sources and may be used as a lightweight evaluation tool.

At the same time, this work does not aim to prevent hallucination or provide guarantees of correctness, and the signals identified should not be used as the sole basis for high-stakes decisions. As with other analysis and detection methods, responsible deployment requires appropriate safeguards and complementary evaluation mechanisms. Overall, we believe the ethical and societal implications of this work are aligned with well-established goals in machine learning research, and do not warrant further specific discussion.

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

# A. A three-step proof that larger $K$ increases (a lower bound on) attention roughness

**Goal.** We consider a *single-layer* causal self-attention mechanism and measure, at a fixed prediction position $t \geq 2$, how *rough* the attention weights over past positions are as a function of the number of mixture components $K$ in the input distribution. Roughness is quantified by the $\ell_2$ adjacent-difference energy

$$R_t \triangleq \sum_{j=1}^{t-2} \left( \alpha_{t,j+1} - \alpha_{t,j} \right)^2, \tag{9}$$

where $\alpha_{t,1:t-1}$ is the attention row (probability vector) used to predict token $t$.

We prove the claim via three sub-claims: (i) adjacent tokens switch mixture components more often when $K$ is larger; (ii) component switches induce larger adjacent differences in *logits* $s_{t,j}$; (iii) softmax transfers logit roughness into attention-weight roughness under mild non-degeneracy.

## A.1. Setup and assumptions

**Mixture of topic labels.** Let $K \in \mathbb{N}$ denote the number of mixture components. Each position $j$ has an associated latent topic label

$$c_j \in \{1, \ldots, K\}.$$

**Assumption A.1** (i.i.d. uniform labels). $(c_j)_{j \geq 1}$ are i.i.d. and $\Pr(c_j = r) = 1/K$ for all $r \in \{1, \ldots, K\}$, which means we randomly assign a topic label for the tokens, and each topic has a unique Gaussian distribution. Then, the input token embedding under the view of Gaussian mixture model will follow a GMM [2]

: For each $j$,

$$x_j = \mu_{c_j} + \varepsilon_j, \qquad \varepsilon_j \overset{i.i.d.}{\sim} \mathcal{N}(0, \sigma^2 I_d),$$

where $\mu_1, \ldots, \mu_K \in \mathbb{R}^d$ are fixed means and $(\varepsilon_j)_j$ are independent of $(c_j)_j$.

Then, if we consider a single-layer causal attention at position $t$ (the position for the next generated token), and Fix $t \geq 2$. Let

$$q_t = W_Q x_t, \qquad k_j = W_K x_j, \quad j \leq t-1,$$

and define logits (pre-softmax scores)

$$s_{t,j} = \frac{q_t^\top k_j}{\sqrt{d_q}} = \frac{q_t^\top W_K x_j}{\sqrt{d_q}}, \qquad j = 1, \ldots, t-1.$$

To simplify the notation system, we further define the induced *score direction* in the input space, where

$$u \triangleq \frac{W_K^\top q_t}{\sqrt{d_q}} \in \mathbb{R}^d, \tag{10}$$

so that

$$s_{t,j} = u^\top x_j. \tag{11}$$

Attention weights from existing token towards next generation token are

$$\alpha_{t,j} = \frac{e^{s_{t,j}}}{\sum_{\ell=1}^{t-1} e^{s_{t,\ell}}}, \qquad j = 1, \ldots, t-1. \tag{12}$$

**Assumption A.2** (Query conditioning). We condition on $x_t$ (equivalently on $q_t$), and treat $u$ in (10) as deterministic. All expectations below are over $\{(c_j, \varepsilon_j)\}_{j \leq t-1}$.

---

[2]Here, we adopt an i.i.d. Gaussian mixture assumption on token embeddings purely for clarity of exposition. The same qualitative scaling result continues to hold under more realistic dependency structures: for example, if latent semantic states follow a hidden Markov model (HMM) with transition matrix $P$, adjacent tokens still switch components with probability $1 - P_{z_t z_t}$, which is strictly positive whenever $P$ is non-degenerate, and the expected attention roughness retains the same scaling in $K$ up to a constant factor reflecting the mixing rate of $P$. We deliberately present the i.i.d. case here because the corresponding HMM derivation introduces substantially heavier notation without altering the underlying mechanism, which would obscure rather than strengthen the intuition we aim to convey.

**Separability along the score direction.**

**Assumption A.3** (Separability). There exists $\Delta > 0$ such that for all $r \neq r'$,

$$|u^\top(\mu_{r'} - \mu_r)| \geq \Delta.$$

In this assumption, we hope that the centroid of Gaussian should be still separable after projection based on $W_k$ and $q_t$ in Equation 10

**Softmax non-degeneracy (prevents vanishing pair mass).** Fix $j \in \{1, \ldots, t-2\}$ and define

$$\Delta s \triangleq s_{t,j+1} - s_{t,j}, \qquad m \triangleq \alpha_{t,j} + \alpha_{t,j+1}.$$

**Assumption A.4** (Softmax non-degeneracy). There exist constants $\eta \in (0,1)$, $B \leq 2$, and $\kappa' > 0$, independent of $K$, such that with

$$E \triangleq \{m \geq \eta\} \cap \{|\Delta s| \leq B\},$$

we have

$$\mathbb{E}\big[(\Delta s)^2 \mathbf{1}_E\big] \geq \kappa' \, \mathbb{E}\big[(\Delta s)^2\big].$$

Intuitively, Assumption A.4 says that local logit fluctuations happen in parts of the sequence that the softmax actually "looks at": with nontrivial frequency, the adjacent pair $(j, j+1)$ receives at least some fixed amount of probability mass, and on those occasions the logit gap is not excessively large. This prevents the situation where logits vary wildly only at positions whose attention weights are almost always zero, in which case attention differences would remain small regardless of logit roughness.

### A.2. Sub-claim 1: Adjacent component switching probability increases with $K$

**Lemma A.5** (Switch probability). *Under Assumption A.1, for any $j \geq 1$, the probability of*

$$\Pr(c_{j+1} \neq c_j) = 1 - \frac{1}{K}.$$

*Proof.* By independence and uniformity,

$$\Pr(c_{j+1} = c_j) = \sum_{r=1}^{K} \Pr(c_j = r, \, c_{j+1} = r) = \sum_{r=1}^{K} \Pr(c_j = r) \Pr(c_{j+1} = r) = \sum_{r=1}^{K} \frac{1}{K} \cdot \frac{1}{K} = \frac{1}{K}.$$

Taking complements gives $\Pr(c_{j+1} \neq c_j) = 1 - \frac{1}{K}$. $\qquad\square$

### A.3. Sub-claim 2: Adjacent logit difference energy increases with $K$

**Lemma A.6** (Logit adjacent-difference energy). *Under Assumptions A.1–A.3 and A.2, for any $j \leq t-2$,*

$$\mathbb{E}\big[(s_{t,j+1} - s_{t,j})^2\big] \geq 2\sigma^2 \|u\|_2^2 + \left(1 - \frac{1}{K}\right)\Delta^2.$$

*Proof.* Using (11) and $x_j = \mu_{c_j} + \varepsilon_j$,

$$s_{t,j+1} - s_{t,j} = u^\top(x_{j+1} - x_j) = u^\top(\mu_{c_{j+1}} - \mu_{c_j}) + u^\top(\varepsilon_{j+1} - \varepsilon_j).$$

Let

$$A \triangleq u^\top(\mu_{c_{j+1}} - \mu_{c_j}), \qquad B \triangleq u^\top(\varepsilon_{j+1} - \varepsilon_j),$$

so that $s_{t,j+1} - s_{t,j} = A + B$. Then

$$\mathbb{E}[(A+B)^2] = \mathbb{E}[A^2] + 2\mathbb{E}[AB] + \mathbb{E}[B^2].$$

The cross term vanishes: $A$ depends only on topic labels $(c_j, c_{j+1})$ while $B$ depends only on Gaussian noises $(\varepsilon_j, \varepsilon_{j+1})$, which are independent of labels, and $\mathbb{E}[B \mid c_j, c_{j+1}] = 0$, hence $\mathbb{E}[AB] = 0$.

For the noise term, $\varepsilon_{j+1} - \varepsilon_j \sim \mathcal{N}(0, 2\sigma^2 I_d)$, so

$$B \sim \mathcal{N}\big(0,\, 2\sigma^2 \|u\|_2^2\big) \quad \Rightarrow \quad \mathbb{E}[B^2] = 2\sigma^2 \|u\|_2^2.$$

For the mean-jump term, if $c_{j+1} = c_j$ then $A = 0$. If $c_{j+1} \neq c_j$, Assumption A.3 gives $A^2 \geq \Delta^2$. Therefore

$$\mathbb{E}[A^2] = \mathbb{E}\big[A^2 \mathbf{1}\{c_{j+1} \neq c_j\}\big] \geq \Delta^2 \, \Pr(c_{j+1} \neq c_j) = \Delta^2\Big(1 - \frac{1}{K}\Big),$$

where we used Lemma A.5. Combining yields the stated lower bound. $\square$

## A.4. Sub-claim 3: Softmax transfers logit roughness to attention roughness

**Lemma A.7** (Pairwise softmax difference identity). *Fix $t$ and $j \leq t - 2$. Let $m = \alpha_{t,j} + \alpha_{t,j+1}$ and $\Delta s = s_{t,j+1} - s_{t,j}$. Then*

$$\alpha_{t,j+1} - \alpha_{t,j} \;=\; m\, \tanh\Big(\frac{\Delta s}{2}\Big).$$

*Proof.* From (12),

$$\alpha_{t,j+1} - \alpha_{t,j} = \frac{e^{s_{t,j+1}} - e^{s_{t,j}}}{Z}, \qquad Z = \sum_{\ell=1}^{t-1} e^{s_{t,\ell}}.$$

Also $m = (e^{s_{t,j+1}} + e^{s_{t,j}})/Z$, hence

$$\alpha_{t,j+1} - \alpha_{t,j} = m \cdot \frac{e^{s_{t,j+1}} - e^{s_{t,j}}}{e^{s_{t,j+1}} + e^{s_{t,j}}} = m\, \tanh\Big(\frac{s_{t,j+1} - s_{t,j}}{2}\Big) = m\, \tanh\Big(\frac{\Delta s}{2}\Big).$$

$\square$

**Lemma A.8** (Softmax transfer lower bound). *Under Assumption A.4 (with $B \leq 2$), for any $j \leq t - 2$,*

$$\mathbb{E}\big[(\alpha_{t,j+1} - \alpha_{t,j})^2\big] \;\geq\; c\, \mathbb{E}\big[(\Delta s)^2\big], \qquad c \triangleq \frac{\eta^2 \kappa'}{16}.$$

*Proof.* By Lemma A.7,

$$(\alpha_{t,j+1} - \alpha_{t,j})^2 = m^2 \tanh^2\Big(\frac{\Delta s}{2}\Big).$$

On $\{|\Delta s| \leq B\}$ with $B \leq 2$, we have $|\Delta s|/2 \leq 1$ and use the elementary bound

$$|\tanh(x)| \geq \frac{|x|}{2} \qquad \text{for all } |x| \leq 1,$$

which implies $\tanh^2(\Delta s/2) \geq (\Delta s)^2/16$ on $\{|\Delta s| \leq B\}$. On $\{m \geq \eta\}$ we have $m^2 \geq \eta^2$. Therefore on the event

$$E = \{m \geq \eta\} \cap \{|\Delta s| \leq B\},$$

$$(\alpha_{t,j+1} - \alpha_{t,j})^2 \geq \eta^2 \cdot \frac{(\Delta s)^2}{16}.$$

Taking expectations and applying Assumption A.4 yields

$$\mathbb{E}\big[(\alpha_{t,j+1} - \alpha_{t,j})^2\big] \geq \frac{\eta^2}{16}\, \mathbb{E}\big[(\Delta s)^2 \mathbf{1}_E\big] \geq \frac{\eta^2 \kappa'}{16}\, \mathbb{E}\big[(\Delta s)^2\big].$$

$\square$

## A.5. Main theorem and concise proof via Sub-claims 1–3

**Theorem A.9** (Monotone $K$-dependent lower bound for attention roughness). *Fix $t \geq 2$ and consider the single-layer causal attention row $\alpha_{t,1:t-1}$ defined in* (12). *Under Assumptions A.1, A.2, A.3, and A.4, there exists a constant $C > 0$ independent of $K$ such that*

$$\mathbb{E}[R_t] \;\geq\; C\,(t-2)\Big(1 - \frac{1}{K}\Big)\Delta^2.$$

*In particular, the right-hand side is monotone increasing in $K$; hence $\mathbb{E}[R_t]$ admits a monotone increasing-in-$K$ lower bound.*

*Proof.* By Lemma A.8, for each $j \leq t - 2$,

$$\mathbb{E}\big[(\alpha_{t,j+1} - \alpha_{t,j})^2\big] \geq c\,\mathbb{E}\big[(s_{t,j+1} - s_{t,j})^2\big], \qquad c = \frac{\eta^2 \kappa'}{16}.$$

Summing over $j = 1, \ldots, t - 2$ gives

$$\mathbb{E}[R_t] = \sum_{j=1}^{t-2} \mathbb{E}\big[(\alpha_{t,j+1} - \alpha_{t,j})^2\big] \geq c \sum_{j=1}^{t-2} \mathbb{E}\big[(s_{t,j+1} - s_{t,j})^2\big].$$

Applying Lemma A.6 yields, for each $j$,

$$\mathbb{E}\big[(s_{t,j+1} - s_{t,j})^2\big] \geq \Big(1 - \frac{1}{K}\Big)\Delta^2,$$

where the factor $\big(1 - \frac{1}{K}\big)$ comes from Lemma A.5. Therefore

$$\mathbb{E}[R_t] \geq c \sum_{j=1}^{t-2} \Big(1 - \frac{1}{K}\Big)\Delta^2 = c\,(t-2)\Big(1 - \frac{1}{K}\Big)\Delta^2.$$

Setting $C = c$ completes the proof. $\qquad\qquad\square$

## A.6. Summary of the logical connection between Sub-claims 1–3

**How the pieces fit together.** The proof decomposes the $K$-dependence into three steps:

  (i) **(Sub-claim 1)** Increasing $K$ increases the probability that adjacent inputs come from different mixture components: $\Pr(c_{j+1} \neq c_j) = 1 - \frac{1}{K}$.

 (ii) **(Sub-claim 2)** Under separability along the attention score direction $u$, a component switch forces a nontrivial expected squared jump in adjacent logits, yielding a lower bound $\mathbb{E}[(s_{t,j+1} - s_{t,j})^2] \geq (1 - \frac{1}{K})\Delta^2$ (up to additional noise energy).

(iii) **(Sub-claim 3)** A pairwise identity for softmax shows $\alpha_{t,j+1} - \alpha_{t,j} = m \tanh(\Delta s/2)$, and under non-degeneracy ($m$ not vanishing too often) this yields $\mathbb{E}[(\alpha_{t,j+1} - \alpha_{t,j})^2] \gtrsim \mathbb{E}[(\Delta s)^2]$.

Chaining (i)–(iii) and summing over $j$ proves Theorem A.9, which provides a monotone increasing-in-$K$ lower bound on $\mathbb{E}[R_t]$.

# B. Frequency-aware Operators' Energy Equivalence

In this appendix section, we provide additional technical details on the spectral operators used in our framework and clarify how high-frequency energy can be consistently quantified using the $\ell_2$ norm. We show that the Discrete Fourier Transform (DFT), Discrete Wavelet Transform (DWT), and the discrete Laplacian operator all provide principled ways to extract high-frequency components from discrete attention signals, and that their corresponding $\ell_2$ norms measure comparable notions of signal energy despite operating in different domains.

## B.1. Discrete Fourier Transform

We begin with the DFT, which provides a global frequency-domain representation of discrete signals. Given a token-level attention signal $a_{l,h} \in \mathbb{R}^T$ from layer $l$ and head $h$, its DFT is defined as

$$\hat{a}_{l,h}(\omega) = \sum_{t=0}^{T-1} a_{l,h}(t)\, e^{-i2\pi\omega t/T}, \quad \omega = 0, \ldots, T-1. \tag{13}$$

A frequency-domain high-pass filter $M(\omega)$ can be applied to isolate high-frequency components, yielding

$$\hat{z}_{l,h}^{\mathrm{hf}}(\omega) = M(\omega)\, \hat{a}_{l,h}(\omega), \tag{14}$$

where $M(\omega) = 1$ for $\omega \in \Omega_{\mathrm{high}}$ and 0 otherwise. The corresponding high-frequency signal in the token domain is obtained via the inverse DFT.

The use of the $\ell_2$ norm as a hallucination score is justified by the *Parseval's theorem* in §3.4, which guarantees energy equivalence between the token and frequency domains:

$$\|z_{l,h}^{\mathrm{hf}}\|_2^2 = \frac{1}{T} \sum_{\omega \in \Omega_{\mathrm{high}}} |\hat{a}_{l,h}(\omega)|^2. \tag{15}$$

Thus, the score $s_{l,h} = \|z_{l,h}^{\mathrm{hf}}\|_2$ directly measures the total high-frequency energy of the attention signal, capturing rapid oscillations and global irregularities across the token sequence.

## B.2. Discrete Wavelet Transform

DWT provides a multi-resolution analysis of discrete signals, decomposing an attention signal into components at different spatial scales. Using an orthonormal wavelet basis (e.g., Daubechies wavelets), the attention signal $a_{l,h}$ is decomposed into approximation coefficients $\mathbf{c_A}$ and detail coefficients $\mathbf{c_D}$ at multiple scales.

In our framework, the high-frequency component $z_{l,h}^{\mathrm{hf}}$ is reconstructed from $\mathbf{c_D}$ corresponding to fine scales, which capture localized and abrupt changes in attention.

By *Parseval's theorem*, the energy of the attention signal is preserved under an orthonormal wavelet transform, and equals the sum of squared wavelet coefficients. Let $d_{j,k}$ denote the detail coefficient $\mathbf{c_D}$ at scale $j$ and position $k$. Then,

$$\|z_{l,h}^{\mathrm{hf}}\|_2^2 = \sum_{j \in \mathcal{J}_{\mathrm{high}}} \sum_k |d_{j,k}|^2, \tag{16}$$

where $\mathcal{J}_{\mathrm{high}}$ denotes the set of high-frequency scales.

Accordingly, the score $s_{l,h} = \|z_{l,h}^{\mathrm{hf}}\|_2$ quantifies the localized detail energy of the attention distribution, emphasizing sharp transitions and spatially concentrated irregularities that are characteristic of unstable attention patterns.

## B.3. Discrete Laplacian Operator

The discrete Laplacian operator provides a spatial-domain alternative for extracting high-frequency variation. For a one-dimensional attention signal $a_{l,h}$, the discrete Laplacian $\mathbf{L}$ computes second-order differences, measuring how much each token's attention deviates from the average of its immediate neighbors.

The $\ell_2$ norm of the Laplacian response,

$$\|\mathbf{L}a_{l,h}\|_2^2, \tag{17}$$

corresponds to the discrete *Dirichlet energy* of the signal, which quantifies its roughness or lack of smoothness (Sandryhaila & Moura, 2014). A higher energy value indicates that the attention distribution exhibits rapid local oscillations or fragmentation, rather than smooth decay or coherent concentration.

The role of the Laplacian as a high-frequency operator can be further understood through its frequency response. In the spectral domain, the discrete Laplacian corresponds to a filter with transfer function

$$H(\omega) = 2 - 2\cos(\omega). \tag{18}$$

As $\omega \to 0$, $H(\omega) \to 0$, suppressing low-frequency (smooth) components. As $\omega \to \pi$, $H(\omega)$ attains its maximum value, strongly amplifying high-frequency components associated with abrupt transitions.

Consequently, the score $s_{l,h} = \|\mathbf{L}a_{l,h}\|_2$ serves as a computationally efficient proxy for high-frequency energy, capturing localized attention instability without requiring an explicit frequency-domain transform.

## C. Experiment Details

### C.1. Implementation of Baselines

#### C.1.1. VERIFICATION AND PROBABILISTIC BASELINES

For verification-based baselines, both **SelfCheckGPT** and **RefChecker** use `gpt-4o-mini` as the backbone model for claim extraction and factual verification, without any supervised training. For probabilistic consistency baselines, **EigenScore** constructs a similarity matrix $\mathbf{S}$ based on BERTScore between generated responses, and derives a consistency score from the largest eigenvalue $\lambda_{\max}(\mathbf{S})$. **ReDeEP** computes two complementary signals during generation: an external context score derived from attention allocation over input tokens, and a parametric knowledge score derived from feed-forward network activations, following the original implementation.

Both EigenScore and ReDeEP are strong unsupervised methods specifically designed for hallucination detection that leverage internal model signals, including attention. Despite not being directly trained for the task, they have shown powerful performance and are therefore included as competitive and fair baselines in our comparison.

#### C.1.2. ATTENTION-BASED FEATURE EXTRACTION

For all intrinsic, attention-based methods (including entropy-based baselines and our frequency-aware features), we extract attention weights from all transformer layers and heads. Let $\mathbf{A} = \{\mathbf{A}_{l,h}\}$ denote the collection of attention distributions, where $\mathbf{A}_{l,h} \in \mathbb{R}^T$ is the token-level attention vector from layer $l$ and head $h$.

Following (Chuang et al., 2024a), we partition attention into context-directed attention $\mathbf{A}_{l,h}^{(c)}$ and generated-token attention $\mathbf{A}_{l,h}^{(g)}$. For a given head, a feature extraction function $f(\cdot)$ is applied to the corresponding attention distribution. Aggregating across all layers and heads yields separate feature vectors for context and generated attention:

$$\mathbf{v}^{(c)} = [f(\mathbf{A}_{1,1}^{(c)}), \ldots, f(\mathbf{A}_{L,H}^{(c)})]^\top, \quad \mathbf{v}^{(g)} = [f(\mathbf{A}_{1,1}^{(g)}), \ldots, f(\mathbf{A}_{L,H}^{(g)})]^\top. \tag{19}$$

The final representation is formed by concatenation, $[\mathbf{v}^{(c)}; \mathbf{v}^{(g)}]$, which is then used for hallucination prediction, consistent with the formulation in the main text.

For entropy-based baselines, the feature function $f(\cdot)$ computes the entropy of the attention distribution for each head:

$$H(\mathbf{A}_{l,h}) = -\sum_i a_i \log a_i, \tag{20}$$

where $a_i$ denotes the normalized attention weight assigned to token $i$.

### C.2. Frequency-aware Operators

**Discrete Fourier Transform.** For DFT, we separate low-frequency and high-frequency components using a frequency cutoff. We systematically vary the cutoff threshold and evaluate hallucination detection performance across models, datasets, and sliding-window settings, as shown in Figure A1. Across most settings, performance improves as the cutoff increases from low values, remains stable over a broad intermediate range, and drops sharply when the cutoff approaches the Nyquist limit.

Here, a normalized frequency of 0.5 corresponds to the Nyquist limit under the DFT formulation. As shown in Figure A1, performance remains stable across a broad range of cutoff frequencies, but begins to degrade when the cutoff exceeds approximately 0.45, with a pronounced drop on the Nyquist boundary. This trend suggests that high-frequency attention variations relevant to hallucination detection are effectively captured below this transition point. Empirically, we find that a cutoff frequency of 0.45 yields consistently strong and stable performance across models and datasets. Additionally, performance is relatively insensitive to lower cutoff settings, with improvements increasing gradually rather than abruptly—indicating robustness to the exact choice of cutoff within a reasonable range.

**Discrete Wavelet Transform.** For DWT, we perform a multi-resolution decomposition of each attention signal $a_{l,h}$ using the Daubechies-4 (db4) wavelet. The db4 wavelet offers a favorable trade-off between locality and smoothness due to its vanishing moments, making it suitable for capturing abrupt transitions in attention distributions. For boundary handling in finite-length attention sequences, we use zero padding for token-level detection and symmetric padding for span-level detection, based on overall performance observed across datasets and models.

We further compare depth-1 (level1) and depth-2 (level2) wavelet decompositions, where the results are shown in Appendix D.3. Empirically, a depth-1 decomposition consistently outperforms deeper decompositions across datasets and models. We attribute this behavior to the fact that higher-level decompositions increasingly mix lower-frequency components into the detail coefficients, reducing their sensitivity to the finest-scale variations that are most informative for hallucination detection. Accordingly, we adopt a level1 wavelet decomposition in all experiments.

**Discrete Laplacian Operator.** The discrete Laplacian requires no cutoff or scale selection. Instead, it directly computes second-order differences in the token domain, acting as a local high-pass filter that amplifies rapid attention fluctuations. This simplicity makes the Laplacian operator computationally efficient and parameter-free, while still capturing localized high-frequency variation in attention distributions. As discussed in Appendix B.3, the $\ell_2$ norm of the Laplacian response corresponds to the discrete Dirichlet energy, providing a principled measure of attention roughness without additional hyperparameters.

### C.3. Model Details

**Inference.** All experiments are conducted on NVIDIA A100 (80GB) GPUs. The LLMs are kept frozen throughout. We perform inference using teacher-forcing decoding with model response tokens to extract attention distributions for spectral analysis.

**Classifiers.** Our method uses a lightweight single-layer Logistic Regression classifier. This choice ensures that the detector does not introduce additional non-linear modeling capacity, and that performance differences primarily reflect the discriminative power of the proposed spectral features, rather than classifier expressiveness. The linear detector was implemented using the `scikit-learn` library. All hyperparameters, including the $\ell_2$ penalty, were kept at their default values, with the exception of the maximum number of iterations, which was set to 1,000 to ensure model convergence.

**Evaluation.** For threshold-dependent metrics such as F1, the decision threshold is selected on a held-out validation split and fixed for test evaluation. In cases where an official validation set is not predefined, we reserve 10% of the original training data for this purpose. We use AUROC as our primary metric because AUROC is threshold-independent and are unaffected by this choice.

### C.4. Dataset Details

We use two publicly available contextual hallucination detection datasets in this work: RAGTruth and HalluRAG.

*Table A1.* Statistics of datasets used in our experiments. Length and ratio are calculated on the token level.

| Dataset | Task | Train | Val | Test | Prompt Len. | Response Len. | Halluc. Ratio |
|---------|------|-------|-----|------|-------------|---------------|---------------|
| RAGTruth | QA | 839 | – | 150 | 439.5 | 208.1 | 0.1045 |
| | D2T | 883 | – | 150 | 892.8 | 225.4 | 0.0768 |
| | Summ | 793 | – | 150 | 840.1 | 153.4 | 0.0527 |
| HalluRAG | QA | 756 | 162 | 162 | 649.3 | 44.4 | 0.1530 |

**RAGTruth** RAGTruth is a large-scale corpus designed for word-level hallucination detection within retrieval-augmented generation (RAG) frameworks (Niu et al., 2024). It consists of 2,965 manually annotated responses with precise hallucination span labels across three primary tasks: question answering (QA), data-to-text (D2T), and summarization (Summ). The dataset provides standardized train and test splits for each task. Specifically, the QA, D2T, and Summ tasks comprise 839, 883, and 793 training samples, respectively, with each task sharing a consistent test set size of 150 samples.

**HalluRAG.** HalluRAG focuses on sentence-level hallucination detection in RAG-based QA scenarios (Ridder & Schilling, 2024). It provides generated responses paired with sentence-level annotations, comprising 756 training, 162 validation, and 162 test samples. Following prior token-level hallucination detection settings, we map each sentence-level label to all constituent tokens during training and evaluation.

More details can be seen in Table A1. All token-level statistics are computed using the tokenizer corresponding to each model, and the reported values are averaged across models. In particular, the hallucination ratio is defined as the number of hallucinated tokens in the response divided by the total number of response tokens. Since this ratio may vary across models, we report the average value over different models.

## D. Full Results for Main Experiments

This section reports the full experimental results and ablation studies omitted from the main text. These results complement the analyses in the main paper and provide additional details on span-level detection, operator configurations, and the robustness of frequency-aware features across different settings.

### D.1. Full Results of Span-level Detection

We report full results for span-level hallucination detection using a sliding-window setting with window size of 8, following the protocol described in the main text. Table A2 presents full performance metrics across models, datasets, and frequency operators.

*Table A2.* Full performance comparison on RagTruth and HalluRAG with chunk size set to 8. For each model, the best result is highlighted in **bold**, and the second-best result is underlined.

| Model / Method | RT-QA | | RT-D2T | | RT-Summ | | HalluRAG | | Overall Avg. | |
|---|---|---|---|---|---|---|---|---|---|---|
| | F1 | AUROC | F1 | AUROC | F1 | AUROC | F1 | AUROC | Avg-F1 | Avg-A |
| **LLaMA-7B** | | | | | | | | | | |
| Lookback-lens | 0.7218 | 0.8467 | 0.7249 | 0.8551 | 0.5852 | 0.6635 | 0.6623 | 0.7856 | 0.6736 | 0.7877 |
| Attn-variance | 0.5077 | 0.6979 | 0.4714 | 0.5886 | 0.4840 | 0.6348 | 0.4446 | 0.5209 | 0.4769 | 0.6106 |
| Attn-entropy | 0.7292 | 0.8502 | 0.7060 | 0.8457 | 0.5489 | 0.6637 | 0.6282 | 0.6909 | 0.6531 | 0.7626 |
| Laplacian | 0.7074 | 0.8365 | 0.7334 | 0.8646 | 0.5833 | 0.6619 | 0.6775 | 0.7754 | 0.6754 | 0.7846 |
| Wavelet-high | 0.7331 | 0.8680 | **0.7478** | 0.8821 | 0.6045 | 0.7199 | 0.6812 | 0.7928 | 0.6917 | 0.8157 |
| Fourier-high | **0.7473** | **0.8725** | 0.7348 | **0.8869** | **0.6188** | **0.7641** | **0.6866** | **0.8100** | **0.6969** | **0.8334** |
| **LLaMA-13B** | | | | | | | | | | |
| Lookback-lens | 0.7004 | 0.8594 | 0.7610 | 0.8872 | 0.5728 | 0.7083 | 0.7096 | 0.8505 | 0.6860 | 0.8264 |
| Attn-variance | 0.5301 | 0.6871 | 0.5329 | 0.7283 | 0.4917 | 0.7014 | 0.5548 | 0.6412 | 0.5274 | 0.6895 |
| Attn-entropy | 0.6876 | 0.8478 | 0.7226 | 0.8602 | 0.5426 | 0.6377 | 0.5872 | 0.6249 | 0.6350 | 0.7427 |
| Laplacian | 0.6966 | 0.8467 | 0.7562 | 0.8853 | 0.5930 | 0.6798 | 0.7217 | 0.8264 | 0.6919 | 0.8096 |
| Wavelet-high | 0.7002 | 0.8685 | 0.7225 | 0.8793 | 0.5821 | 0.7202 | **0.7417** | 0.8438 | 0.6866 | 0.8279 |
| Fourier-high | **0.7365** | **0.8863** | **0.7706** | **0.8988** | **0.6119** | **0.7904** | 0.7217 | **0.8515** | **0.7102** | **0.8568** |
| **Mistral-7B** | | | | | | | | | | |
| Lookback-lens | 0.7920 | **0.9206** | 0.7751 | 0.9002 | 0.6992 | 0.8082 | 0.7752 | 0.8403 | 0.7604 | 0.8673 |
| Attn-variance | 0.7036 | 0.8216 | 0.6339 | 0.8112 | 0.4741 | 0.6507 | 0.5709 | 0.7215 | 0.5956 | 0.7512 |
| Attn-entropy | 0.7857 | 0.9007 | 0.7300 | 0.8656 | 0.6606 | 0.7727 | 0.6597 | 0.7140 | 0.7090 | 0.8132 |
| Laplacian | 0.7500 | 0.8822 | 0.7719 | 0.8928 | 0.6747 | 0.7734 | **0.8056** | **0.8920** | 0.7506 | 0.8601 |
| Wavelet-high | 0.7839 | 0.9083 | 0.7287 | 0.8866 | 0.6771 | 0.8073 | 0.7682 | 0.8802 | 0.7395 | 0.8706 |
| Fourier-high | **0.8042** | 0.9190 | **0.7833** | **0.9057** | **0.7114** | **0.8188** | 0.7883 | 0.8872 | **0.7718** | **0.8827** |

We additionally analyze the effect of span window size in span-level hallucination detection. Following prior work, the default setting uses a sliding window size of 8. We vary the window size from 4 to 32 and report AUROC on RAGTruth using Fourier-high features with LLaMA-7B.

Performance peaks at window size 8, consistent with prior span-level hallucination detection methods. Larger windows progressively smooth local instability signals, making hallucination boundaries less distinguishable.

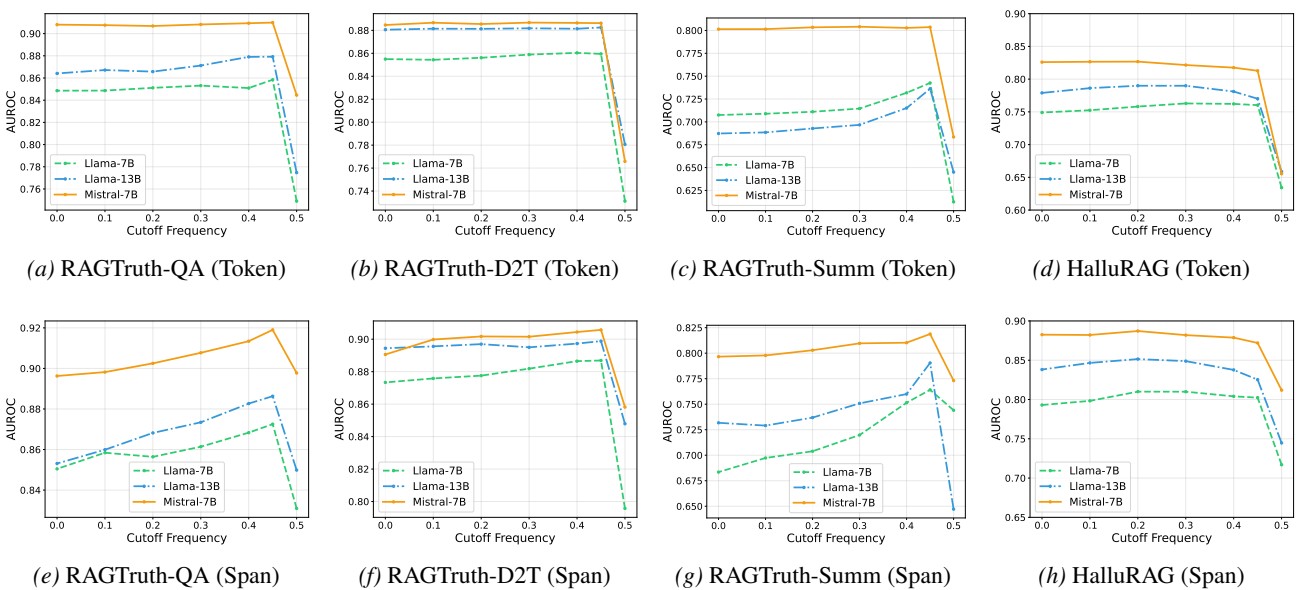

*(a)* RAGTruth-QA (Token)  *(b)* RAGTruth-D2T (Token)  *(c)* RAGTruth-Summ (Token)  *(d)* HalluRAG (Token)

*(e)* RAGTruth-QA (Span)  *(f)* RAGTruth-D2T (Span)  *(g)* RAGTruth-Summ (Span)  *(h)* HalluRAG (Span)

*Figure A1.* Ablation study on different frequency cutoffs at token and span levels on hallucination detection performance. The top row (a-d) shows results at the token level, while the bottom row (e-h) shows results at the span level.

*Table A3.* Effect of span window size on span-level hallucination detection (AUROC).

| Window Size | RT-QA | RT-D2T | RT-Summ | Avg |
|---|---|---|---|---|
| 4 | 0.8604 | 0.8716 | 0.7089 | 0.8136 |
| 8 | **0.8725** | **0.8869** | **0.7641** | **0.8412** |
| 16 | 0.8400 | 0.8728 | 0.6561 | 0.7897 |
| 32 | 0.8252 | 0.8556 | 0.6391 | 0.7733 |

## D.2. Full Results of DFT Operator

We report the full results for hallucination detection using the Discrete Fourier Transform (DFT) with different frequency cutoffs, as shown in Figure A1.

## D.3. Full Results of DWT Operator

We report the full results for hallucination detection using the Discrete Wavelet Transform (DWT), comparing level-1 and level-2 wavelet decompositions, as shown in Table A4 and Table A5.

*Table A4.* Performance comparison on token level using Wavelet-high across levels and padding methods. Best results within each model are in **bold**; second-best are underlined.

| Model/Level | Padding | RT-QA | | RT-D2T | | RT-Summ | | HalluRAG | | Overall Avg. | |
|---|---|---|---|---|---|---|---|---|---|---|---|
| | | F1 | AUROC | F1 | AUROC | F1 | AUROC | F1 | AUROC | Avg-F1 | Avg-A |
| **LLaMA-7B** | | | | | | | | | | | |
| | zero | 0.7194 | 0.8526 | **0.6898** | **0.8569** | **0.5929** | 0.7165 | **0.6384** | **0.7550** | **0.6601** | **0.7953** |
| level1 | period | 0.7146 | 0.8514 | 0.6887 | 0.8517 | 0.5799 | 0.7091 | 0.6360 | 0.7449 | 0.6548 | 0.7893 |
| | symm | **0.7215** | **0.8557** | 0.6875 | 0.8545 | 0.5880 | **0.7182** | 0.6353 | 0.7521 | 0.6581 | 0.7951 |
| | zero | 0.7157 | 0.8522 | 0.6893 | 0.8531 | 0.5822 | 0.7088 | 0.6359 | 0.7464 | 0.6558 | 0.7901 |
| level2 | period | 0.7155 | 0.8514 | 0.6891 | 0.8528 | 0.5814 | 0.7069 | 0.6344 | 0.7438 | 0.6551 | 0.7887 |
| | symm | 0.7170 | 0.8519 | 0.6879 | 0.8531 | 0.5851 | 0.7159 | 0.6375 | 0.7543 | 0.6569 | 0.7938 |
| **LLaMA-13B** | | | | | | | | | | | |
| | zero | **0.7029** | **0.8741** | **0.7383** | **0.8932** | 0.5651 | 0.7042 | 0.6684 | 0.7809 | 0.6687 | **0.8131** |
| level1 | period | 0.6962 | 0.8679 | 0.7207 | 0.8811 | 0.5678 | 0.7044 | 0.6527 | 0.7675 | 0.6593 | 0.8052 |
| | symm | 0.7002 | 0.8685 | 0.7225 | 0.8793 | **0.5821** | **0.7202** | **0.6718** | **0.7839** | **0.6691** | 0.8130 |
| | zero | 0.6971 | 0.8711 | 0.7213 | 0.8815 | 0.5699 | 0.7043 | 0.6567 | 0.7701 | 0.6613 | 0.8068 |
| level2 | period | 0.6945 | 0.8675 | 0.7199 | 0.8816 | 0.5687 | 0.7045 | 0.6515 | 0.7669 | 0.6587 | 0.8051 |
| | symm | 0.6993 | 0.8701 | 0.7216 | 0.8834 | 0.5678 | 0.7046 | 0.6710 | 0.7789 | 0.6649 | 0.8093 |
| **Mistral-7B** | | | | | | | | | | | |
| | zero | **0.7876** | **0.9117** | 0.7136 | 0.8829 | 0.6849 | **0.8075** | 0.7274 | 0.8360 | 0.7284 | 0.8595 |
| level1 | period | 0.7812 | 0.9077 | 0.7228 | 0.8830 | 0.6840 | 0.8034 | 0.7139 | 0.8229 | 0.7255 | 0.8542 |
| | symm | 0.7839 | 0.9083 | **0.7287** | 0.8866 | 0.6771 | 0.8073 | 0.7130 | 0.8231 | 0.7257 | 0.8563 |
| | zero | 0.7873 | 0.9113 | 0.7238 | 0.8826 | 0.6844 | 0.8038 | 0.7213 | 0.8265 | 0.7292 | 0.8561 |
| level2 | period | 0.7770 | 0.9059 | 0.7238 | 0.8826 | 0.6829 | 0.8041 | 0.7130 | 0.8228 | 0.7242 | 0.8539 |
| | symm | 0.7831 | 0.9104 | 0.7265 | **0.8869** | **0.6884** | 0.8069 | **0.7308** | **0.8413** | **0.7322** | **0.8614** |

*Table A5.* Performance comparison when chunk size is 8 using Wavelet-high across levels and padding methods. Best results within each model are in **bold**; second-best are underlined.

| Model/Level | Padding | RT-QA | | RT-D2T | | RT-Summ | | HalluRAG | | Overall Avg. | |
|---|---|---|---|---|---|---|---|---|---|---|---|
| | | F1 | AUROC | F1 | AUROC | F1 | AUROC | F1 | AUROC | Avg-F1 | Avg-A |
| **LLaMA-7B** | | | | | | | | | | | |
| | zero | 0.7313 | 0.8611 | **0.7479** | 0.8785 | 0.6020 | 0.7169 | **0.6846** | **0.8047** | 0.6915 | 0.8153 |
| level1 | period | 0.7271 | 0.8572 | 0.7426 | 0.8748 | 0.5973 | 0.7008 | 0.6786 | 0.7963 | 0.6864 | 0.8073 |
| | symm | **0.7331** | **0.8680** | 0.7478 | **0.8821** | 0.6045 | **0.7199** | 0.6812 | 0.7928 | **0.6917** | **0.8157** |
| | zero | 0.7277 | 0.8594 | 0.7461 | 0.8760 | 0.5996 | 0.6997 | 0.6800 | 0.7953 | 0.6884 | 0.8076 |
| level2 | period | 0.7272 | 0.8579 | 0.7452 | 0.8752 | 0.5975 | 0.6972 | 0.6792 | 0.7937 | 0.6873 | 0.8060 |
| | symm | 0.7277 | 0.8602 | 0.7478 | 0.8768 | **0.6049** | 0.7107 | 0.6771 | 0.7938 | 0.6894 | 0.8104 |
| **LLaMA-13B** | | | | | | | | | | | |
| | zero | 0.7050 | 0.8616 | **0.7833** | **0.9091** | 0.5974 | 0.7557 | 0.7306 | 0.8479 | 0.7041 | 0.8436 |
| level1 | period | 0.7105 | 0.8610 | 0.7634 | 0.8951 | 0.5984 | 0.7471 | 0.7141 | 0.8338 | 0.6966 | 0.8343 |
| | symm | **0.7200** | **0.8718** | 0.7616 | 0.8928 | **0.6043** | **0.7697** | **0.7482** | **0.8541** | **0.7085** | **0.8471** |
| | zero | 0.7079 | 0.8612 | 0.7638 | 0.8953 | 0.5940 | 0.7499 | 0.7168 | 0.8351 | 0.6956 | 0.8354 |
| level2 | period | 0.7102 | 0.8617 | 0.7637 | 0.8957 | 0.5957 | 0.7485 | 0.7163 | 0.8337 | 0.6965 | 0.8349 |
| | symm | 0.7094 | 0.8625 | 0.7664 | 0.8962 | 0.5967 | 0.7563 | 0.7274 | 0.8474 | 0.7000 | 0.8406 |
| **Mistral-7B** | | | | | | | | | | | |
| | zero | **0.7984** | 0.9111 | 0.7686 | **0.9080** | 0.7010 | 0.8111 | 0.7904 | 0.8777 | 0.7646 | 0.8770 |
| level1 | period | 0.7858 | 0.9108 | **0.7875** | 0.8997 | 0.7027 | 0.8084 | 0.7900 | 0.8717 | 0.7665 | 0.8726 |
| | symm | 0.7970 | 0.9052 | 0.7820 | 0.8995 | 0.7001 | **0.8179** | 0.7921 | 0.8841 | 0.7678 | 0.8767 |
| | zero | 0.7879 | **0.9127** | 0.7887 | 0.8995 | **0.7037** | 0.8096 | 0.7902 | 0.8729 | 0.7676 | 0.8737 |
| level2 | period | 0.7865 | 0.9116 | 0.7887 | 0.8995 | 0.7023 | 0.8079 | 0.7896 | 0.8727 | 0.7668 | 0.8729 |
| | symm | 0.7919 | 0.9074 | 0.7837 | 0.9000 | 0.7035 | 0.8128 | **0.8063** | **0.8876** | **0.7713** | **0.8770** |

## D.4. Full Results of Ablation Study

This section reports the full results corresponding to the attention-based analyses discussed in the main paper, evaluated across all datasets and models studied.

Figure A2 reports the full results comparing low-pass and high-pass Fourier attention features across all evaluated models, datasets, and both token-level and span-level settings.

Figure A3 shows the complete layer-wise importance profiles of frequency-aware attention features across models, including both token-level and span-level detection.

Table A6 presents the full results of the head-level sparsity analysis, reporting detection performance when restricting attention features to the Top-$k$ most important heads across datasets and models.

Table A7 provides the complete ablation results comparing context-only and generated-only attention features for different spectral operators across all evaluation settings.

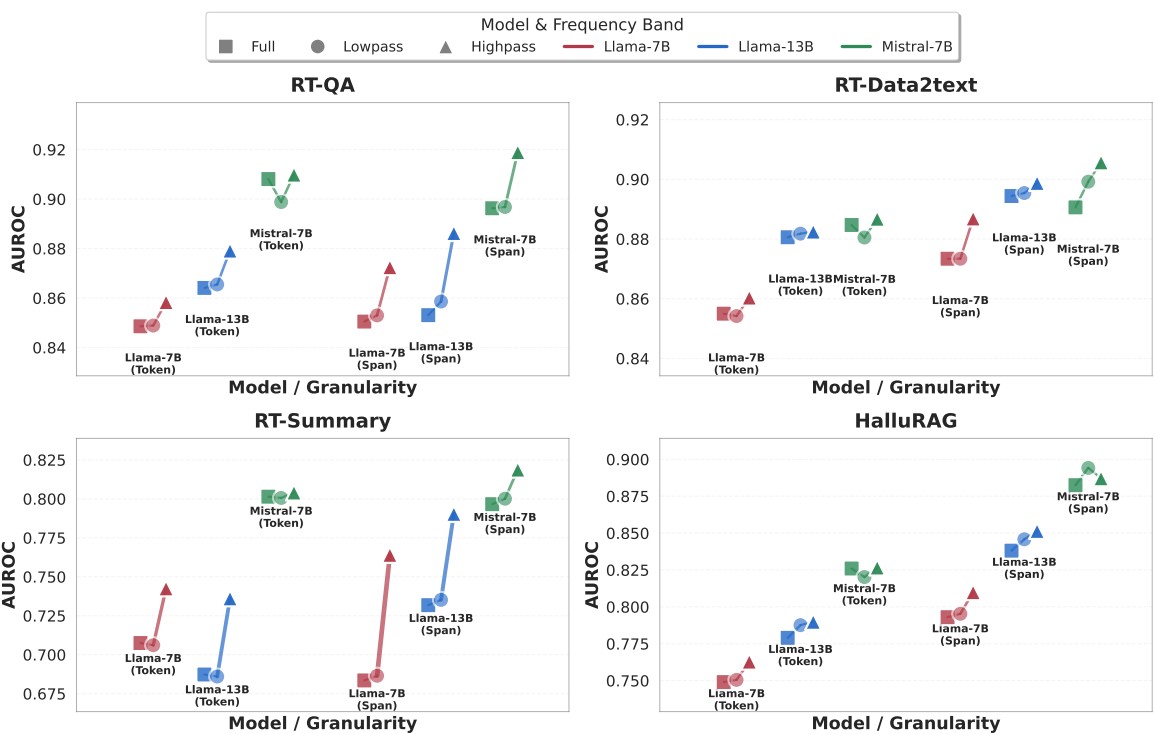

*Figure A2.* Full results of comparing Full-, Low-, and High-Pass Fourier attention features. Average AUROC across models under token- and span-level evaluation settings.

## D.5. Examples of Raw Attention Signals

Figure A4 presents qualitative examples of raw attention distributions from individual attention heads. Each row corresponds to the same attention head at a fixed layer, while the left and right columns show attention over a hallucinated token and a non-hallucinated token, respectively.

We emphasize that these attention distributions reflect real model behavior and are substantially more irregular than the schematic examples shown in Figure 2. In practice, attention weights are often sparse, unevenly distributed, and exhibit non-trivial fluctuations across token positions, rather than forming unimodal patterns (Nawrot et al., 2025).

Notably, for the examples shown in the top row, the two tokens share identical lookback ratios as measured by Lookback-Lens. Similarly, for the bottom row, the context entropy of the two attention distributions is equal. In these cases, these aggregate

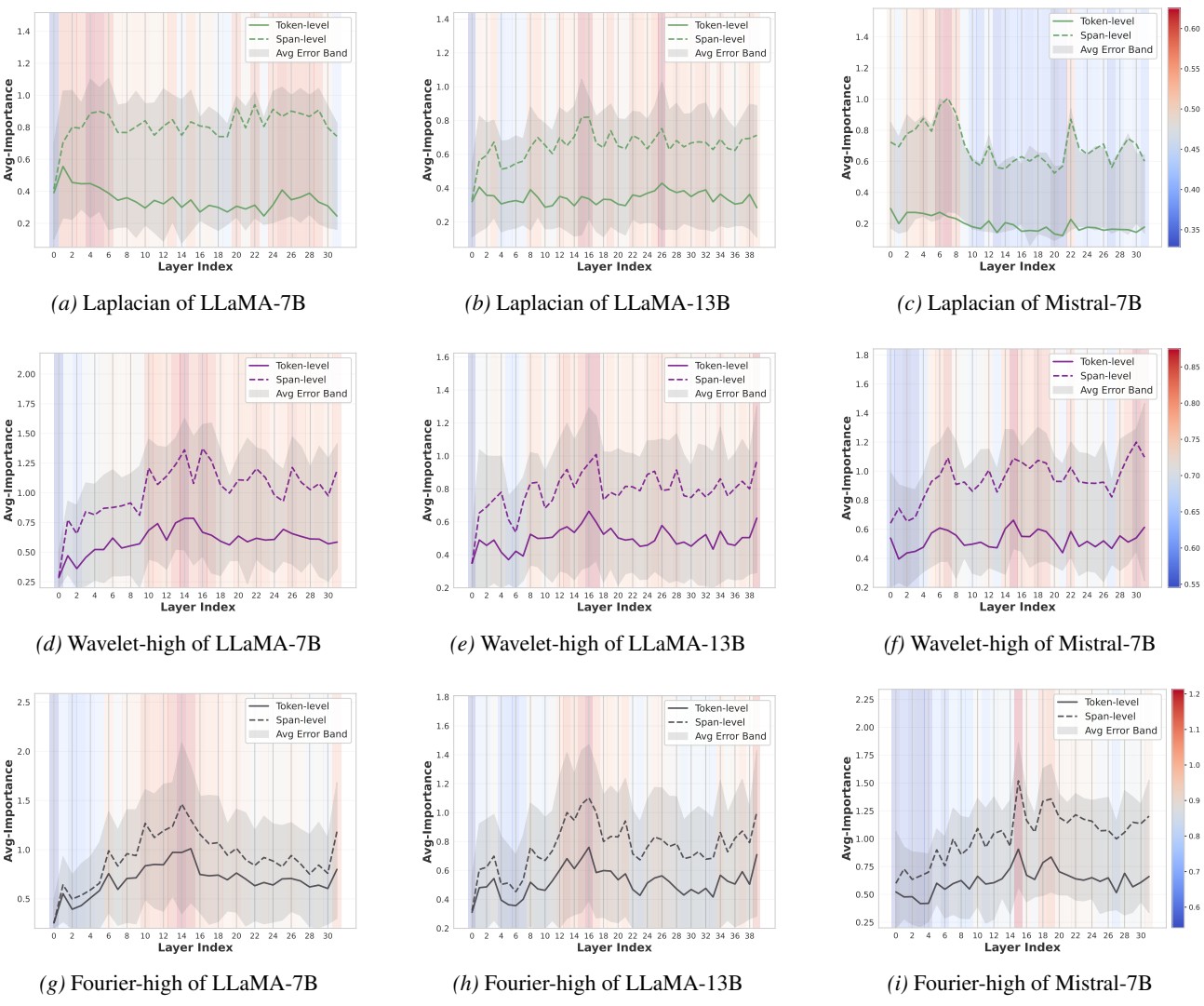

*Figure A3.* Full results for layer-wise importance. Solid and dashed lines correspond to token-level and span-level detection, respectively. The shaded region indicates the standard deviation of head-level importance within each layer.

*Table A6.* Original vs. Top-$k$ head only Performance.

(a) AUROC Results for LLaMA-13B.

| | | RagTruth-Avg | | |
|---|---|---|---|---|
| **Method** | **Original** | **Top-$k$ heads** | | |
| | | *k = 100* | *50* | *10* |
| Laplacian | 0.8016 | 0.7858 | 0.7595 | 0.6424 |
| Wavelet-high | 0.8238 | 0.8079 | 0.7821 | 0.7191 |
| Fourier | 0.8326 | 0.8204 | 0.7899 | 0.7041 |
| | | **HalluRAG** | | |
| **Method** | **Original** | **Top-$k$ heads** | | |
| | | *k = 100* | *50* | *10* |
| Laplacian | 0.7624 | 0.6924 | 0.6764 | 0.6335 |
| Wavelet-high | 0.7809 | 0.7433 | 0.7133 | 0.6292 |
| Fourier-high | 0.7899 | 0.7749 | 0.7572 | 0.6455 |

(b) AUROC Results for Mistral-7b.

| | | RagTruth-Avg | | |
|---|---|---|---|---|
| **Method** | **Original** | **Top-$k$ heads** | | |
| | | *k = 100* | *50* | *10* |
| Laplacian | 0.8625 | 0.8312 | 0.7839 | 0.6795 |
| Wavelet-high | 0.8673 | 0.8483 | 0.8317 | 0.7707 |
| Fourier-high | 0.8669 | 0.8440 | 0.8283 | 0.7624 |
| | | **HalluRAG** | | |
| **Method** | **Original** | **Top-$k$ heads** | | |
| | | *k = 100* | *50* | *10* |
| Laplacian | 0.8098 | 0.7611 | 0.7433 | 0.6807 |
| Wavelet-high | 0.8360 | 0.7715 | 0.7467 | 0.7091 |
| Fourier-high | 0.8267 | 0.7855 | 0.7587 | 0.6920 |

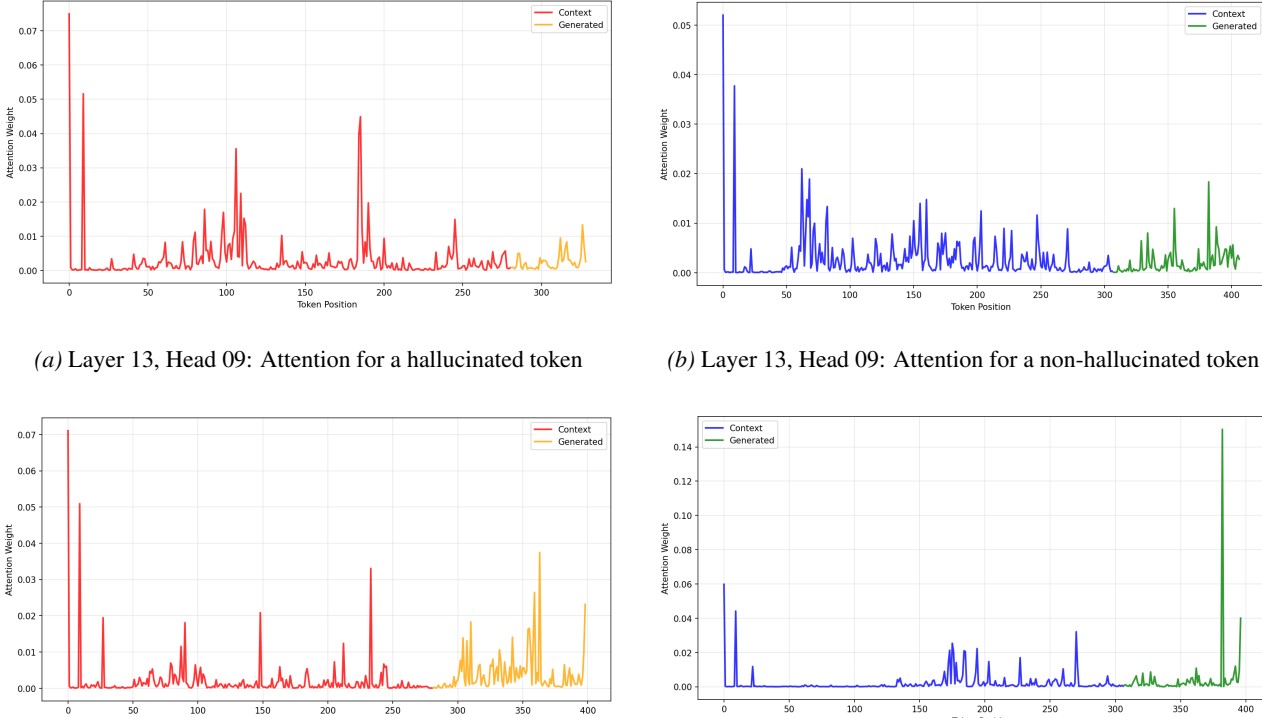

*(a)* Layer 13, Head 09: Attention for a hallucinated token

*(b)* Layer 13, Head 09: Attention for a non-hallucinated token

*(c)* Layer 15, Head 29: Attention for a hallucinated token

*(d)* Layer 15, Head 29: Attention for a non-hallucinated token

*Figure A4.* Raw attention signal visualizations. Each subfigure compares raw attention distributions for a hallucinated token (left) and a non-hallucinated token (right). Red and yellow curves denote attention over context and generated tokens, respectively, when generating a hallucinated token, while blue and green curves denote attention over context and generated tokens for a non-hallucinated token.

*Table A7.* Ablation study comparing context-only and generated-only attention features. Results are reported across spectral operators and models.

| Model | Setting | Operator | RagTruth-Avg | | HalluRAG | |
|---|---|---|---|---|---|---|
| | | | F1 | AUROC | F1 | AUROC |
| LLaMA-7B | Original | Laplacian | 0.6588 | 0.8003 | 0.6370 | 0.7429 |
| | | Wavelet-high | 0.6673 | 0.8087 | 0.6384 | 0.7550 |
| | | Fourier-high | 0.6685 | 0.8205 | 0.6478 | 0.7629 |
| | Context-only | Laplacian | 0.6504 | 0.8014 | 0.6340 | 0.7396 |
| | | Wavelet-high | 0.6541 | 0.8074 | 0.6402 | 0.7479 |
| | | Fourier-high | 0.6544 | 0.8138 | 0.6385 | 0.7538 |
| | Generated-only | Laplacian | 0.6441 | 0.7855 | 0.6183 | 0.7240 |
| | | Wavelet-high | 0.6430 | 0.7889 | 0.6264 | 0.7334 |
| | | Fourier-high | 0.6416 | 0.7939 | 0.6262 | 0.7349 |
| LLaMA-13B | Original | Laplacian | 0.6526 | 0.8016 | 0.6659 | 0.7624 |
| | | Wavelet-high | 0.6688 | 0.8238 | 0.6684 | 0.7809 |
| | | Fourier-high | 0.6759 | 0.8326 | 0.6732 | 0.7899 |
| | Context-only | Laplacian | 0.6603 | 0.8144 | 0.6538 | 0.7627 |
| | | Wavelet-high | 0.6634 | 0.8206 | 0.6524 | 0.7697 |
| | | Fourier-high | 0.6688 | 0.8333 | 0.6581 | 0.7815 |
| | Generated-only | Laplacian | 0.6468 | 0.7988 | 0.6430 | 0.7503 |
| | | Wavelet-high | 0.6579 | 0.8080 | 0.6524 | 0.7697 |
| | | Fourier-high | 0.6446 | 0.8053 | 0.6473 | 0.7642 |
| Mistral-7B | Original | Laplacian | 0.7262 | 0.8625 | 0.7001 | 0.8098 |
| | | Wavelet-high | 0.7287 | 0.8673 | 0.7274 | 0.8360 |
| | | Fourier-high | 0.7300 | 0.8669 | 0.7221 | 0.8267 |
| | Context-only | Laplacian | 0.7289 | 0.8636 | 0.7207 | 0.8242 |
| | | Wavelet-high | 0.7280 | 0.8655 | 0.7176 | 0.8238 |
| | | Fourier-high | 0.7299 | 0.8668 | 0.7128 | 0.8227 |
| | Generated-only | Laplacian | 0.7151 | 0.8515 | 0.6840 | 0.7802 |
| | | Wavelet-high | 0.7185 | 0.8548 | 0.6994 | 0.7983 |
| | | Fourier-high | 0.7178 | 0.8531 | 0.6895 | 0.7913 |

statistics alone are insufficient to distinguish hallucinated from non-hallucinated cases by visual inspection. In contrast, after applying a high-pass filter to the same attention signals, the resulting high-frequency energy differs substantially between the two cases. Although the raw attention curves may appear similar at a coarse level, frequency-domain filtering reveals differences in fine-grained variation patterns.

# E. Additional Robustness Analysis

To further evaluate the robustness of our frequency-aware attention features, we conduct additional analyses on attention sinks, cross-model generalization, long-context settings, and multi-hop reasoning task.

## E.1. Robustness to Attention Sinks

A potential concern is whether the proposed frequency-aware features are dominated by the well-known attention sink phenomenon (Gu et al., 2025a), where excessively large attention mass is allocated to initial tokens. Since attention sinks introduce a strong component near the beginning of the sequence, they may potentially affect the frequency spectrum.

To evaluate this effect, we remove attention of the initial sink tokens and re-evaluate the detectors. Table A8 reports AUROC before and after sink removal on LLaMA-7B.

*Table A8.* AUROC before and after removing attention sinks on LLaMA-7B.

| Method | RT-QA | RT-D2T | RT-Summ |
|---|---|---|---|
| Lookback-Lens | $0.848 \rightarrow 0.765$ | $0.844 \rightarrow 0.787$ | $0.716 \rightarrow 0.769$ |
| Fourier-high | $0.858 \rightarrow 0.765$ | $0.860 \rightarrow 0.801$ | $0.743 \rightarrow 0.776$ |

Both methods exhibit similar performance changes after sink removal, suggesting that attention sinks do not unduly contribute to the effectiveness of our frequency-aware features. Instead, the discriminative signals primarily arise from localized high-frequency variations beyond the sink region.

## E.2. Generalization Across Model Families

To additionally examine whether the proposed frequency-aware features generalize across fundamentally different tokenizer architectures, we evaluate the method on Qwen2.5-7B-Instruct using the same normalized Fourier-high cutoff frequency ($0.45$) without additional tuning. The results are reported in Table A9.

*Table A9.* AUROC on Qwen2.5-7B-Instruct using the same Fourier cutoff frequency ($0.45$).

| Method | RT-QA | RT-D2T | RT-Summ | Avg |
|---|---|---|---|---|
| Lookback-Lens | 0.605 | 0.751 | 0.514 | 0.623 |
| Laplacian | 0.606 | 0.760 | 0.583 | 0.650 |
| Wavelet-high | **0.615** | 0.759 | 0.575 | 0.650 |
| Fourier-high | 0.614 | **0.760** | **0.594** | **0.656** |

The results show that frequency-aware attention features remain effective across substantially different tokenizer architectures. In particular, Fourier-high achieves the strongest average performance without requiring tokenizer-specific tuning, suggesting that the proposed high-frequency signals capture model behaviors that generalize beyond a specific tokenization scheme.

## E.3. Long-Context Robustness

We further evaluate the robustness of frequency-aware attention features under substantially longer contexts using Long-BenchV2 (Bai et al., 2025). Since Qwen2.5-7B-Instruct supports substantially longer context windows (up to 128K tokens [3]) than the LLaMA and Mistral models used in our main experiments, we adopt it for long-context evaluation and test on two LongBenchV2 domains with average context lengths up to 49k tokens.

Although all methods become more challenging as attention becomes increasingly spread across extremely long contexts, frequency-aware features consistently outperform Lookback-Lens across both domains. This suggests that high-frequency attention variation remains informative even when attention distributions are broadly dispersed over long sequences.

---

[3]https://huggingface.co/Qwen/Qwen2.5-7B-Instruct

*Table A10.* AUROC on LongBenchV2 using Qwen2.5-7B-Instruct.

| Method | Government (20k) | Financial (49k) |
|---|---|---|
| Lookback-Lens | 0.604 | 0.349 |
| Laplacian | 0.775 | 0.515 |
| Wavelet-high | 0.792 | 0.591 |
| Fourier-high | 0.792 | 0.546 |

### E.4. Multi-hop Reasoning vs. Hallucination in Attention

Another subtle question is whether the proposed high-frequency features might incorrectly treat legitimate reasoning behavior as hallucination. In particular, valid multi-hop reasoning often requires attention to shift across multiple distant evidence spans, which may also produce non-trivial attention transitions during generation. To investigate whether such structured reasoning behavior will be confused with hallucination-induced instability, we evaluate our detectors on the multi-hop QA benchmarks HotpotQA and MuSiQue using Qwen2.5-7B-Instruct.

We focus on the subset of examples with exact-match (EM) correct answers ($EM = 1$), where the generated answers are fully supported by the input. Under this setting, any token predicted as hallucinated can be treated as a misjudgment of valid reasoning behavior.

*Table A11.* Hallucination detector predictions on correct multi-hop reasoning answers.

| Method | HotpotQA | | MuSiQue | |
| | Total Flagged | Misjudged | Total Flagged | Misjudged |
|---|---|---|---|---|
| Lookback-Lens | 2482 | 3/14306 (0.02%) | 1912 | 1/2704 (0.04%) |
| Laplacian | 1142 | 2/14306 (0.01%) | 749 | 2/2704 (0.07%) |
| Wavelet-high | 1213 | 3/14306 (0.02%) | 909 | 1/2704 (0.04%) |
| Fourier-high | 2706 | 4/14306 (0.03%) | 1632 | 1/2704 (0.04%) |

Across both datasets, the misjudgment rate remains below $0.07\%$, indicating that frequency-aware detectors rarely penalize valid multi-hop reasoning despite actively identifying hallucination signals.

## F. Additional Analysis on Transferability and Supervision

*Table A12.* Cross-domain evaluation. Models are trained on the column domain (Source) and evaluated on the row domain (Target). **Bold** values indicate in-domain performance (diagonal, Target=Source). Darker shading corresponds to higher cross-domain performance.

| Method | Target | Token-Level (Source) | | | Span-Level (Source) | | |
| | | QA | D2T | Summ | QA | D2T | Summ |
|---|---|---|---|---|---|---|---|
| Lookback-lens | QA | **0.8482** | 0.7839 | 0.7741 | **0.8467** | 0.8140 | 0.7985 |
| | D2T | 0.6340 | **0.8442** | 0.7211 | 0.6334 | **0.8551** | 0.7275 |
| | Summ | 0.6618 | 0.6400 | **0.7156** | 0.6563 | 0.6496 | **0.6635** |
| Laplacian | QA | **0.8449** | 0.7928 | 0.7881 | **0.8365** | 0.7898 | 0.7857 |
| | D2T | 0.6438 | **0.8519** | 0.6809 | 0.6147 | **0.8646** | 0.6264 |
| | Summ | 0.6428 | 0.6678 | **0.7040** | 0.6132 | 0.6596 | **0.6619** |
| Wavelet-high | QA | **0.8526** | 0.8056 | 0.8076 | **0.8680** | 0.8316 | 0.8418 |
| | D2T | 0.6525 | **0.8569** | 0.7179 | 0.6405 | **0.8821** | 0.7307 |
| | Summ | 0.6630 | 0.6773 | **0.7165** | 0.6541 | 0.7032 | **0.7199** |
| Fourier-high | QA | **0.8584** | 0.8282 | 0.8233 | **0.8725** | 0.8561 | 0.8593 |
| | D2T | 0.6668 | **0.8595** | 0.7228 | 0.6434 | **0.8869** | 0.7584 |
| | Summ | 0.6781 | 0.7040 | **0.7426** | 0.6748 | 0.7296 | **0.7641** |

## F.1. Cross-domain Transfer Analysis

To examine whether the detector captures intrinsic attention-based signals rather than overfits task-specific artifacts, we conduct cross-domain transfer evaluation, where detectors are trained on one task domain and evaluated on another, as shown in Table A12. This experiment also serves to evaluate the generalization ability of the classifier and to test its susceptibility to overfitting.

Overall, spectral-based detectors exhibit more stable cross-domain behavior compared to Lookback-Lens. In particular, Fourier- and Wavelet-based variants maintain stronger performance when transferring across task boundaries, whereas Lookback-Lens shows larger performance degradation under domain shift. This suggests that frequency-domain attention features capture more task-robust signals than heuristics based on attention mass or recency.

We also observe an asymmetric transfer pattern across tasks. Models trained on QA generally transfer worse to Data-to-Text and Summarization than models trained on Data-to-Text or Summarization transferring to QA. This asymmetry holds consistently across methods and detection granularities. A plausible explanation is that QA exhibits more constrained and localized attention patterns, which may limit the generality of learned aggregation weights when applied to structurally different generation tasks.

## F.2. Transfer Across Benchmarks

To evaluate whether the learned detector generalizes beyond within-benchmark settings, we train detectors on HalluRAG and directly evaluate on RAGTruth without re-fitting, as shown in Table A13.

*Table A13.* Cross-benchmark transfer from HalluRAG to RAGTruth (AUROC).

| Method | RT-QA | RT-D2T | RT-Summ | Avg |
|---|---|---|---|---|
| Lookback-Lens | 0.5535 | 0.6024 | 0.5587 | 0.5715 |
| Laplacian | 0.5812 | **0.6128** | **0.5635** | 0.5858 |
| Wavelet-high | 0.5772 | 0.5833 | 0.5442 | 0.5682 |
| Fourier-high | **0.6689** | 0.5951 | 0.5599 | **0.6080** |

Despite substantial distribution shifts between benchmarks, Fourier-high achieves the strongest average performance, suggesting that high-frequency attention variation captures relatively transferable signals of ungrounded generation.

## F.3. Unsupervised Instantiations

As discussed in Appendix C.1.1, several baselines considered in this work operate without supervised labels. To enable a more direct comparison with these methods, we evaluate two instantiations of our frequency-aware features without ground-truth labels.

**No-Label Instantiation.** For unsupervised settings, standard clustering methods such as K-Means are less suitable in this setting due to the high feature dimensionality (e.g., 2048 for LLaMA-7B) and the strong class imbalance, where hallucinated tokens constitute only a small minority. Instead, hallucination detection is better viewed as an outlier detection problem, where hallucinated tokens correspond to sparse abnormal patterns that do not form coherent clusters. We therefore adopt Local Outlier Factor (LOF), a KNN-based unsupervised method that identifies outliers by comparing the local density of each point to that of its neighbors.

We first apply PCA ($d = 30$) for dimensionality reduction, followed by LOF with $n_{\text{neighbors}} = 150$ and contamination rate 0.05.

*Table A14.* Unsupervised hallucination detection using LOF (AUROC).

| Method | RT-QA | RT-D2T | RT-Summ | Avg |
|---|---|---|---|---|
| EigenScore | 0.5253 | 0.5297 | 0.4989 | 0.5180 |
| ReDeEP | 0.6364 | 0.3960 | 0.5760 | 0.5361 |
| Laplacian+LOF | 0.6678 | 0.5587 | 0.5823 | 0.6029 |
| Wavelet+LOF | **0.6684** | **0.5632** | **0.5858** | **0.6058** |
| Fourier+LOF | 0.6621 | 0.5483 | 0.5417 | 0.5840 |

The results suggest that frequency-aware attention features alone already encode meaningful hallucination-related structure, even without supervised calibration.

**Pseudo-Label Instantiation.**   To provide a fairer comparison with verification-based methods, we additionally replace human-annotated hallucination labels with pseudo-labels generated by `gpt-4o-mini`, matching the backbone used by SelfCheckGPT and RefChecker.

Under this setting, the detector remains free of human supervision, as pseudo-labels are only used for lightweight calibration rather than manual annotation. We further include a direct prompt-based LLM-as-a-judge baseline for comparison.

*Table A15.* Pseudo-label calibration using `gpt-4o-mini` (AUROC).

| Method | RT-QA | RT-D2T | RT-Summ | Avg |
|---|---|---|---|---|
| SelfCheckGPT | 0.6942 | 0.8026 | 0.6674 | 0.7214 |
| RefChecker | 0.5865 | 0.6349 | 0.6080 | 0.6098 |
| Direct Prompt | 0.6864 | 0.6932 | **0.7155** | 0.6984 |
| Laplacian | **0.8191** | **0.8300** | 0.7029 | **0.7840** |
| Wavelet-high | 0.8021 | 0.8218 | 0.6924 | 0.7721 |
| Fourier-high | 0.7924 | 0.8179 | 0.6842 | 0.7648 |

The results indicate that frequency-aware attention features capture complementary intrinsic signals beyond those provided by LLM-as-a-judge verification alone.

### F.4. Effect of Classifier Capacity

To verify that the gains mainly originate from the proposed features rather than classifier complexity, we compare logistic regression with MLP classifiers.

*Table A16.* Comparison between logistic regression and MLP classifiers (AUROC).

| Classifier | RT-QA | RT-D2T | RT-Summ | Avg |
|---|---|---|---|---|
| Logistic Regression | **0.8584** | 0.8595 | **0.7426** | **0.8202** |
| MLP (hidden=64) | 0.8502 | 0.8538 | 0.7225 | 0.8088 |
| MLP (hidden=128) | 0.8474 | **0.8621** | 0.7276 | 0.8124 |

All MLP classifiers were trained to convergence. As shown in Table A16, increasing classifier capacity with 64- or 128-dimensional hidden layers does not consistently improve over logistic regression, suggesting that the discriminative power primarily originates from the frequency-aware attention features themselves rather than the downstream classifier.

## G. Analysis on Rare Entity Tokens

High-frequency features can be used to characterize attention fluctuations during generation. One possible concern is that such high-frequency attention patterns may primarily reflect rare tokens or named entities, rather than hallucinations themselves. To examine this issue, we conduct a fine-grained analysis across named-entity categories using SpaCy[4] NER annotations.

More specifically, We rank tokens in the LLaMA-2-7B-Chat vocabulary by frequency and define the bottom 25% as rare tokens. For each sample, we compute the proportion of rare tokens in the context and split RAGTruth into High-Rarity and Low-Rarity subsets using the per-task median. We then analyze performance across fine-grained named-entity categories.

We report the performance gap $\Delta$ between the High-Rarity and Low-Rarity subsets, defined as: $\Delta = \text{AUROC}_{\text{High}} - \text{AUROC}_{\text{Low}}$. Positive values indicate better performance on high-rarity samples, while negative values indicate performance degradation under high-rarity contexts.

Across all 16 categories, including numerical and temporal expressions, frequency-aware features exhibit no excessive sensitivity to rarity relative to existing attention-based methods. Overall, these results suggest that the proposed frequency-

---

[4]https://spacy.io/

*Table A17.* Performance comparison across high-rarity and low-rarity subsets for different named-entity categories. Darker colors correspond to larger absolute performance gaps ($\Delta$).

| NER Type | % in NER | LBLens$_{High}$ | LBLens$_{Low}$ | LBLens$_\Delta$ | Fourier$_{High}$ | Fourier$_{Low}$ | Fourier$_\Delta$ | Example |
|---|---|---|---|---|---|---|---|---|
| PERSON | 5.73% | 0.8068 | 0.8184 | −0.0116 | 0.8240 | 0.8164 | +0.0076 | Steve Austin |
| NORP | 0.88% | 0.8076 | 0.7981 | +0.0094 | 0.8222 | 0.7872 | +0.0351 | Greek |
| FAC | 0.29% | 0.8196 | 0.7887 | +0.0309 | 0.8253 | 0.7826 | +0.0427 | Kennedy Center |
| ORG | 9.26% | 0.8116 | 0.8149 | −0.0033 | 0.8173 | 0.8217 | −0.0043 | Medicaid |
| GPE | 3.24% | 0.8292 | 0.8383 | −0.0091 | 0.8244 | 0.8148 | +0.0096 | Virginia |
| LOC | 0.28% | 0.8173 | 0.7829 | +0.0344 | 0.8303 | 0.7834 | +0.0469 | Caribbean Sea |
| PRODUCT | 2.05% | 0.8312 | 0.7684 | +0.0628 | 0.8798 | 0.8217 | +0.0581 | DoorDash |
| EVENT | 0.23% | 0.8072 | 0.7741 | +0.0331 | 0.7749 | 0.7192 | +0.0557 | Civil War |
| WORK_OF_ART | 0.91% | 0.8139 | 0.7752 | +0.0387 | 0.8059 | 0.7964 | +0.0095 | Anarchy Workout |
| CARDINAL | 31.85% | 0.8138 | 0.7828 | +0.0310 | 0.8324 | 0.8061 | +0.0263 | 970 |
| ORDINAL | 0.42% | 0.8259 | 0.7777 | +0.0483 | 0.7809 | 0.7637 | +0.0172 | 14th |
| QUANTITY | 5.02% | 0.8310 | 0.7851 | +0.0459 | 0.8259 | 0.7743 | +0.0516 | 1 cup |
| PERCENT | 4.57% | 0.8262 | 0.7710 | +0.0471 | 0.8209 | 0.7821 | +0.0388 | 25% |
| MONEY | 3.62% | 0.8220 | 0.7952 | +0.0268 | 0.8356 | 0.8063 | +0.0293 | £250 |
| DATE | 25.34% | 0.8226 | 0.8070 | +0.0156 | 0.8199 | 0.8204 | −0.0005 | July 2003 |
| TIME | 6.31% | 0.8242 | 0.7822 | +0.0421 | 0.8466 | 0.7930 | +0.0537 | 1 pm |

aware features are not merely driven by rare entities or numerical expressions, but instead capture broader hallucination-related attention dynamics.

