# OpenReview forum: "Detecting Contextual Hallucinations in Large Language Models with Frequency-Aware Attention"
_ICML.cc/2026/Conference — ICML 2026 regular_

### Official Review · Reviewer_hZpP · 2026-03-04

**Soundness:** 4
**Presentation:** 3
**Significance:** 3
**Originality:** 3
**Overall Recommendation:** 4
**Confidence:** 3

**Summary:**

The paper addresses the problem of context-based token-level hallucination detection. They observe empirically that when attention distributions across tokens are viewed as discrete signals, hallucinated tokens more often correspond to high-frequency variations. Motivated by this observation, the authors formalize the problem of hallucination detection using frequency analysis on attention variation. They then experimentally test their approach against hallucination detection baselines, and analyse it to gain additional insight on how different layers, heads, frequency components and context/generated tokens influence the results.

**Compliance With Llm Reviewing Policy:**

Affirmed.

**Final Justification:**

I maintain my score, as my main concern (being more explicit about the method's limitations) is addressed with the addition of a limitation section. I believe the paper presents a promising new tool for hallucination detection that does however have meaningful limitations.

**Key Questions For Authors:**

1. I am having some trouble understanding Figures 4 and 5. Specifically, in Figure 4, you report the average AUROC. What is this average over? Did you aggregate all tasks into one average? If that is the case, I would recommend just showing one task in the main and the rest in the appendix, to not obscure the result. If you are averaging over something else, then please elaborate (and see if it is appropriate to add error bars). Secondly, in Figure 5, what is the colorbar showing? Can you add a label to it?

2. Some parts of the setup are a bit unclear to me. In particular, can you explain in section 5.4.3 how you computed the importance? And in 3.4, how did you set K_high?

3. Can you comment on how computationally (in)efficient your method is compared to the baselines you consider?

4. There was a recent study [1] showing that many hallucination detection methods highly correlate with output length (Table 5), though that study focused on output-level hallucination detection as opposed to token-level, as you did. I wonder if you observed something similar in your experiments, ie whether hallucinated tokens appeared generally later in the output or more often in longer outputs.


[1] [The Illusion of Progress: Re-evaluating Hallucination Detection in LLMs](https://aclanthology.org/2025.emnlp-main.1761/) (Janiak et al., EMNLP 2025)

**Limitations:**

No. I believe a discussion/limitations section would add to the paper. As mentioned in Weaknesses, the authors could comment on practical barriers on adopting this method, its computational efficiency, any assumptions particular to their settings, as well as the scope of their evaluation.

**Strengths And Weaknesses:**

**STRENGTHS**

- The idea of the paper is interesting and presented in a well written manner, making the paper enjoyable to read. I also appreciated the related work sections, they situate the work clearly in the broader hallucination detection area and in relation to the most closely related method.

- The frequency-based approach consistently outperforms baseline methods and also applies to a longer token-span setting.

- The analysis of how various components contribute to hallucination was well-thought and provided interesting insight.


**WEAKNESSES**

- The paper lacks discussion about the limitations of the method. Some comments about the practicability of adopting the method, the evaluation scope, the computational cost, any important assumptions about the applicability in different settings etc would be greatly appreciated.

- The computational and memory cost of the method is not discussed and seems nontrivial as it requires extracting attention and doing DFT across all layers and heads per token, which is more demanding than the baselines.

- The method relies on attention being accessible, which is not the case for users of proprietary models or API-based settings, limiting its applicability.

- Additionally the method requires training a supervised linear classifier on token-level hallucination data, which is not easily available. While the authors show that results transfer across domains relatively better compared to existing approaches, there is still some performance degradation when training and testing on different tasks, especially if the tasks are not similarly structured, as the authors show.

**Minor**

- Typo in line 184 left "that indexed by"

---

> ### Author Rebuttal · Authors · 2026-03-31
>
> We thank the reviewer for the thorough and constructive feedback. We address each question below.
>
>
> ## Q1. Figures 4 and 5 clarification.
>
> In Figure 4, the Average AUROC is computed across all datasets (RT-QA, -D2T, -Summ, HalluRAG). Thanks for your suggestion, we will show one representative task in the main text, with the full results in appendix.
> In Figure 5, the colorbar represents the same average importance metric as the left y-axis, with shading added for clearer layer-wise visualization. We will revise the figure to avoid confusion.
>
> ## Q2. Importance computation (§5.4.3) and K_high setting.
>
> The importance in §5.4.3 is measured by the average absolute coefficient of the logistic regression classifier for each feature dimension (see Line368-369, we will make it more explicit). Each dimension corresponds to one (layer, head, context/generated attention) triple, so we can aggregate by context/generated attention (Figure 6) or by layer (Figure 5) to analyze their relative contributions.
>
> For K_high (§3.4, Eq. 5), we set the normalized frequency threshold to 0.45 and retain frequency indices above it. This value was selected based on the analysis in Figure A1, which shows stable performance across a broad range.
> We will state this more explicitly in the revised text.
>
> ## Q3. Computational efficiency.
>
> Our method shares the same attention extraction step as other attention-based baselines (e.g., LookbackLens). The additional overhead lies mainly in frequency-domain transforms. To provide a concrete comparison, we report per-sample inference time on RAGTruth-QA (NVIDIA A100,LLaMA-7B):
>
> **Table R6: Efficiency**
>
> | Method | Per Sample (ms) |
> |---|---|
> | EigenScore | 3916.2 |
> | SelfCheckGPT | 3371.8 |
> | Lookback-Lens | 1846.5 |
> | Wavelet-high | 3085.2 |
> | Fourier-high | 2432.4 |
>
> While our method introduces moderate overhead compared to Lookback-Lens due to the frequency-domain transforms, it remains substantially faster than verification-based methods.
>
>
> ## Q4. Correlation between hallucination and output length.
>
> To assess sensitivity to output length, we group test samples into three equal-sized bins by response length and compute both token-level hallucination rate and AUROC.
>
> **Table R8: Performance on Different Output Length ( LLaMA-7B, RAGTruth)**
>
> | Task | Group | Token Range | Halluc. Rate (%) | Fourier | Lookback-Lens|
> |:---:|:---:|:---:|:---:|:---:|:---:|
> | QA | Short | ≤193 | 6.10 | 0.733 | 0.725 |
> | QA | Medium | 194–271 | 5.64 | 0.899 | 0.898 |
> | QA | Long | >271 | 9.20 | 0.886 | 0.864 |
> | D2T | Short | ≤172 | 5.25 | 0.804 | 0.775 |
> | D2T | Medium | 173–200 | 4.88 | 0.840 | 0.824 |
> | D2T | Long | >200 | 7.79 | 0.886 | 0.874 |
> | Summ | Short | ≤115 | 6.19 | 0.766 | 0.751 |
> | Summ | Medium | 116–163 | 2.74 | 0.817 | 0.809 |
> | Summ | Long | >163 | 4.87 | 0.692 | 0.675 |
>
> In QA and D2T, the Long group exhibits the highest hallucination rate, while the Medium group has the lowest hallucination rate. Despite this non-uniform distribution of hallucinations across length, Fourier maintains stable AUROC and outperforms Lookback-Lens across all settings. This indicates that our frequency-aware features capture genuine attention instability rather than relying on output length as a shortcut signal.
>
>
>
> ## Limitations
>
> Here we address the limitations identified by the reviewer. We will add a dedicated Limitations section in the revised paper.
>
> **Computational cost.**
> While individually lightweight, applying frequency transforms across all layers, heads, and tokens adds overhead beyond scalar-based methods (e.g., lookback ratio). We will include detailed runtime comparisons in the revision.
>
> **Applicability to closed-source models.**
> Our method requires access to internal attention weights, which are unavailable in API-based model. A potential solution is to use an open-source model as a proxy: given the (context, response) pair from a closed-source model, one could apply teacher-forcing on an open-source model and extract its attention patterns.
> The intuition is that if the response contains hallucinations, the mismatch between the generated text and the context may also induce unstable attention patterns in the proxy model.
> This idea is related to works on using open-source models as proxies for closed-source systems. However, distributional differences between open- and closed-source models may still limit the reliability of this approach, and we leave empirical validation to future work.
>
> **Supervised training data.**
> Our detector requires hallucination annotations to train the classifier. While our cross-domain transfer results (Table A2) demonstrate that frequency-aware features generalize better than baselines across task boundaries, performance still degrades when training and testing on structurally dissimilar tasks. Reducing dependence on task-specific labeled data remains an important direction for future work.
>
> ## Minor
>
> Many thanks! We will correct this typo.

---

> > ### Author Rebuttal · Reviewer_hZpP · 2026-04-03
> >
> > I thank the authors for their response. My main concern was that several limitations of the method were not made explicit, but this is resolved with the addition of a limitations section, that the authors committed to add. I will keep my score.

---

> > > ### Author Response · Authors · 2026-04-08
> > >
> > > Thank you very much for your positive feedback, and for highlighting the importance of clearly discussing the limitations.
> > > We are glad that our responses have helped address your concerns, and we will include a dedicated limitations section in the updated version to ensure greater transparency and completeness.

---

### Official Review · Reviewer_j1jJ · 2026-03-10

**Soundness:** 3
**Presentation:** 3
**Significance:** 3
**Originality:** 3
**Overall Recommendation:** 4
**Confidence:** 2

**Summary:**

This paper addresses the illusion problem that occurs when LLMs generate based on context, and proposes a frequency-aware attention analysis method. It treats the attention distribution of each token during the generation process as a discrete signal, and uses frequency analysis to depict the structural characteristics of attention changes with the position of the token.

**Compliance With Llm Reviewing Policy:**

Affirmed.

**Final Justification:**

This paper proposes an interesting and original frequency-aware perspective for contextual hallucination detection, and the empirical results are generally strong. I found the method intuitive, the analysis thorough, and the paper clearly written.

My main concerns were about the reliance on teacher-forcing and the practical overhead of frequency-domain analysis during inference. The rebuttal addressed these concerns well by providing additional autoregressive evaluation, efficiency comparisons, and clarification that the performance gains mainly come from the proposed features rather than classifier complexity.

Some limitations still remain, especially regarding applicability to closed-source/API-only models and the dependence on labeled data for the final detector. Overall, however, I think the paper makes a meaningful contribution, and I maintain my original evaluation.

**Key Questions For Authors:**

- The paper employed simple logistic regression. If a slightly more complex nonlinear classifier (such as MLP) were used instead, would the performance be significantly improved? And is this improvement attributed to the features themselves or the model's capacity?
- Although it is called lightweight, will performing DFT/DWT transformations in real time during long text inference significantly increase the latency of the first character or the processing pressure?

**Limitations:**

Yes

**Strengths And Weaknesses:**

Strengths：

- Theoretical support is solid. The paper not only presents intuitive experimental observations, but also uses Parseval's theorem to prove the equivalence of time-domain and frequency-domain energy, providing a solid foundation for feature extraction.
- The experimental design is comprehensive. It compared verification type (SelfCheckGPT), internal representation type (EigenScore), and various attention baselines, covering models of sizes ranging from 7B to 13B.

Weaknesses：

- The method mainly relies on teacher-forcing to extract the attention distribution, which differs from the generation process during actual reasoning. This may affect the effectiveness of the detection method in online scenarios.
- During the reasoning process, each layer and each head needs to perform DFT or DWT transformation in real time. In scenarios of large-scale concurrency or long context (long text), this token-by-token frequency domain calculation may accumulate into a significant delay that cannot be ignored.

---

> ### Author Rebuttal · Authors · 2026-03-31
>
> We thank the reviewer for the thoughtful and constructive feedback. Below we address each concern.
> ## W1: Teacher-forcing vs. autoregressive decoding
>
> This is an important point. First, we note that teacher-forcing is a standard protocol in prior intrinsic detection work (e.g., Lookback-Lens, ReDeEP) for controlled and reproducible evaluation. Under teacher-forcing, the model processes the same token sequence as in actual generation, preserving the attention distributions that the model would produce during normal decoding.
>
> To further validate our method under autoregressive decoding, we conducted additional experiments on Qwen2.5-7B-Instruct. Specifically, we used it to autoregressively generate responses for the RAGTruth test set, extracted attention distributions during generation, and annotated hallucinations using GPT-4.1 with human verification. The results below confirm that our frequency-aware features have consistent performance under this autoregressive generation setting:
>
>
> **Table R2: Qwen2.5-7B (AUROC)**
>
> | Method | RT-QA | RT-D2T | RT-Summ | Avg |
> |--------|-------|--------|---------|-----|
> | Lookback-Lens | 0.605 | 0.751 | 0.514 | 0.623 |
> | Laplacian | 0.606 | 0.760 | 0.583 | 0.650 |
> | Wavelet-high | **0.615** | 0.759 | 0.575 | 0.650 |
> | Fourier-high | 0.614 | **0.760** | **0.594** | **0.656** |
>
>
>
> ## W2 & Q2: Time latency of DFT/DWT
>
> We appreciate this practical concern. We compare our method against unsupervised baselines (SelfCheckGPT, EigenScore) and attention-based methods (Lookback-Lens) in per-sample end-to-end inference time on RAGTruth-QA (NVIDIA A100):
>
>
> **Table R6: Efficiency**
>
> | Method | Per Sample (ms) |
> |---|---|
> | EigenScore | 3916.2 |
> | SelfCheckGPT | 3371.8 |
> | Lookback-Lens | 1846.5 |
> | Wavelet-high (ours) | 3085.2 |
> | Fourier-high (ours) | 2432.4 |
>
>
> All single-pass attention-based methods (Lookback-Lens, Fourier-high, Wavelet-high) share the same model forward pass for attention extraction. Compared to Lookback-Lens, which computes only a scalar ratio per head, our methods add moderate overhead for the spectral feature extraction step, while remaining substantially faster than SelfCheckGPT and EigenScore, which require multiple sampled generations.
>
>
>
> ## Q1: Would a nonlinear classifier (e.g., MLP) improve performance?
>
> To verify that performance gains stem from the features rather than classifier capacity, we compared logistic regression with MLPs on LLaMA-7B (Fourier-high, token-level AUROC):
>
> **Table R7: MLP Classification**
>
> | Classifier | RT-QA | RT-D2T | RT-Summ | Avg |
> |---|---|---|---|---|
> | Logistic Regression | **0.8584** | 0.8595 | **0.7426** | **0.8202** |
> | MLP (hidden=64) | 0.8502 | 0.8538 | 0.7225 | 0.8088 |
> | MLP (hidden=128) | 0.8474 | **0.8621** | 0.7276 | 0.8124 |
>
>
> Note that MLP classifiers were trained to convergence. From the results, MLPs do not improve over logistic regression with 64 and 128 hidden dimensions, which confirms that the discriminative power for hallucination detection lies mainly in the frequency-aware features themselves rather than the classifier.
>
> Additionally, the linear classifier is not merely a simplification but a design choice: its learned coefficients directly support the interpretability analyses in §5.4, including layer-wise importance, head-wise sparsity, and the relative contribution of context vs. generated attention, which are central to understanding why frequency-aware features are effective.
>
> We sincerely thank the reviewer for the insightful questions and constructive suggestions, which have helped strengthen our paper.

---

> > ### Author Rebuttal · Reviewer_j1jJ · 2026-04-03
> >
> > I have no further questions. I find the current score appropriate and retain my original evaluation.

---

> > > ### Author Response · Authors · 2026-04-08
> > >
> > > Thank you again for your insightful and constructive feedback.
> > >
> > > We are glad that our clarifications have addressed your concerns, and we will include the additional experiments and analysis in the final version to further improve the paper.

---

### Official Review · Reviewer_bic9 · 2026-03-11

**Soundness:** 3
**Presentation:** 4
**Significance:** 3
**Originality:** 4
**Overall Recommendation:** 5
**Confidence:** 4

**Summary:**

# The problem:
Hallucinations are a well-known and significant issue in LLMs. It is useful to be able to detect hallucinations during generation at a token and span level. This paper focus on one very specific type of hallucination that occurs when additional non-grounded information is added in a generation during a Q/A setting.

# The solution:
The authors propose a frequency-aware view attention where attention over prior tokens is treated as a 1D discrete signal. Then, a high-pass operator is applied to isolate high-frequency components and then a logistic regression is trained on the resulting features to classify hallucination vs non-hallucinations.

# The insights:
The main insight is that hallucinated tokens (specifically in the context of un-grounded information) is associated with high-frequency sparse attention energy. Also, the paper shows that the most useful signal for this lies in middle layers and is concentrated in a relatively small subset of heads.

**Compliance With Llm Reviewing Policy:**

Affirmed.

**Final Justification:**

I believe the idea behind this paper is interesting and novel. I originally had concerns about the empirical evaluation being robust and sound enough to support the authors' claims. After the rebuttal process, the authors have addressed the concerns I raised above satisfactorily. I am improving my score to a 5 (accept).

**Key Questions For Authors:**

Nothing aside from what's already mentioned above in the weaknesses and nits

**Limitations:**

I believe the authors could do a better job of discussing the limitations of their method in relation to unsupervised baselines.

**Strengths And Weaknesses:**

# Strengths

The paper addresses an important problem with clear motivation. The authors explain why model-internal signals can be useful, and why coarse attention summaries might miss local information.

The core methodology proposed by the authors is novel, and the method is interpretable and lightweight.

The empirical section of the paper is strong. The authors evaluate across multiple models, tasks, etc. I also appreciated the useful studies on layer importance, operator choice, etc.

The paper is overall well written and easy to follow. The figures are well made and explanatory.

# Weaknesses

The evaluation setup is not perfectly apples-to-apples; a lot of the baselines (SelfCheckGPT, RefChecker, etc) are essentially unsupervised / zero-shot. The method proposed by the paper requires training a supervised regression. The comparisons are still informative, but the paper should be more explicit about the mismatch. Ideally, I would like the authors to call more attention to cross-domain transfer evaluation in the main paper (instead of the appendix), and potentially also try to see how transfer across benchmarks or models would work in order to understand the potential to use this as a more general hallucination detection tool during model deployment.

Separately, I believe that high frequency sparse attention patterns would also correlate with rare entities, numbers, etc. Has there been additional analysis to understand if hallucinations are the *only* thing that causes the attention pattern described in the paper's hypothesis?

I believe these two missing / underexplored areas are important for the paper. If addressed, I believe the paper has the potential to be quite strong.

# Nits:

At line 256 you say HalluRag has token-level annotation but on line 983 you say it has sentence level annotation.

---

> ### Author Rebuttal · Authors · 2026-03-31
>
> We sincerely thank the reviewer for the constructive feedback. Below we address each concern.
>
> ## W1: Baseline Comparison Setup
>
> **Supervised vs Unsupervised.**
> Indeed, in Table 1, SelfCheckGPT, RefChecker, EigenScore, and ReDeEP are unsupervised baselines. We have (partially) noted this in Appendix C1.1. We will make this explicit in the main text during the revision.
>
> We acknowledge that the reviewer says the comparisons are still informative. We include these baselines because they cover the two major detection paradigms: SelfCheckGPT and RefChecker represent post-hoc verification powered by GPT-4o-mini, providing a strong LLM-as-a-judge reference, while EigenScore and ReDeEP exploit complementary model-internal signals (probabilities, hidden states, attention mass), enabling a direct comparison of which intrinsic signals are most informative.
>
> However, we would like to note that the core frequency-aware part of our method is entirely supervision-free and derived from attention dynamics. Supervision is only used at the final stage to calibrate a decision rule. In this sense, our method is closer to unsupervised feature extraction + minimal supervised calibration, rather than a fully supervised approach.
>
> As is also shown in Table R7 below, replacing the logistic regression by MLP for the final part won’t lead to overall better performance. Thus, one possible reason is that the discriminative power for hallucination lies mainly in the unsupervised part.
>
> **Table R7: MLP Classification (AUROC)**
>
> | Classifier | RT-QA | RT-D2T | RT-Summ | Avg |
> |---|---|---|---|---|
> | Logistic Regression | 0.8584 | 0.8595 | 0.7426 | 0.8202 |
> | MLP (hidden=64) | 0.8502 | 0.8538 | 0.7225 | 0.8088 |
> | MLP (hidden=128) | 0.8474 | 0.8621 | 0.7276 | 0.8124 |
>
> That said, our method does leverage supervision to achieve improved discrimination, and we will clarify this point in the revision.
>
> **Cross-domain transfer evaluation**
> Thanks for your suggestion. We will elevate the cross-domain transfer analysis (Appendix D.1, including Table A2) and will move it into the main text in the next version. As shown there, our frequency-based features exhibit more robust cross-domain transfer than Lookback-Lens.
>
> We attribute the superior cross-domain transfer performance of our method to the nature of the signal. Lookback-Lens relies on the attention ratio between the context and previously generated tokens, which can vary substantially across tasks due to differences in generation length and context structure. In contrast, our method focuses on high-frequency attention variation, which reflects local instability within the context. This provides a more task-invariant signal of ungrounded generation, regardless of the overall allocation pattern.
>
> ---
>
> ## W2: Do Rare Entities Also Affect High-Frequency Attention?
>
> This is an excellent question. To investigate this, we conducted an additional analysis examining whether context rarity confounds our detector's performance.
>
> We measure context rarity by computing the proportion of infrequent entity tokens in each sample's context. Specifically, we rank tokens in LLaMA-2-7B-Chat's vocabulary by their frequency and label the bottom 25% as rare tokens.
> For each test sample, we calculate the rare-token ratio in its context and split samples at the per-task median into High-Rarity (N=75) and Low-Rarity (N=75) groups.
>
> **Table R5: Results (AUROC, LLaMA-7B):**
>
> | Method | Task | High-Rarity | Low-Rarity | Δ |
> |:--|:--|:--:|:--:|:--:|
> | Lookback-Lens | RT-QA | 0.8611 | 0.8266 | +0.035 |
> | | RT-D2T | 0.8529 | 0.8340 | +0.019 |
> | | RT-Summ | 0.7040 | 0.7237 | −0.020 |
> | | **Avg** | 0.8060 | 0.7948 | 0.024 (abs) |
> | Fourier-high | RT-QA | 0.8764 | 0.8204 | +0.056 |
> | | RT-D2T | 0.8650 | 0.8540 | +0.011 |
> | | RT-Summ | 0.7462 | 0.7401 | +0.006 |
> | | **Avg** | 0.8292 | 0.8048| 0.024 (abs) |
>
> From Table R5, we observe that context rarity does not disproportionately affect our method, suggesting our frequency-aware features do not confuse rarity with hallucination more than existing methods. Notably, for Summarization, where rare entities are less prevalent, Fourier-high shows near-zero sensitivity to rarity (Δ = +0.006), while maintaining advantage over Lookback-Lens.
>
> ---
>
> ## Typo: HalluRAG Annotation Granularity
>
> Thank you for spotting this. HalluRAG provides sentence-level annotations. We map these to the token level by assigning sentence-level labels to all constituent tokens, which we acknowledge is a coarser supervision signal than RAGTruth's native token-level spans. We will clarify this in the revised text to resolve the ambiguity.

---

> > ### Author Rebuttal · Reviewer_bic9 · 2026-04-02
> >
> > Thank you for the clarifications and the additional analysis on rarity!
> >
> > 1. The new analysis on rarity is indeed helpful. However, my original query also included numbers -- Intuitively, I would expect this would also have spiky attention since I would imagine it would reference a single token from the context (e.g. if the context is "There are 12 puppies in the basket", then a question about how many objects are in the basket would have a lot of attention over the token for "12"). I would expect the same for dates (like "The person was born on 04/02/1900"). The reason I am asking is because I think it is very important for the main claim of the paper that the authors provide a very strong analysis addressing anything else that could potentially cause this same behavior other than hallucinations.
> >
> > 2.  Thank you for the clarifications here. On the supervised/unsupervised setting, I think my original concern about the baseline mismatch was not fully understood. I will try to restate it more precisely. My point was not that the feature extraction stage has to be supervised, or that the classifier used is too expressive. The detector still uses hallucination labels to fit a decision rule. Thus, from the lens of trying to deploy this as a hallucination detection model, it is a **supervised** detector. While Appendix D.1 provides useful within RAGTruth cross-task transfer results, I would still like to understand whether this learned rule transfers across benchmarks, or if there as a no-label instantiation that can work. I was previously suggesting one or both of the following experiments:
> > -  How well do the rules learned on hallucination labels from HalluRAG perform on RAGTruth?
> > - Is there a no-label instantiation of the method? For example, could you use a clustering algorithm instead to see if there's a strong separation between the hallucinations vs non-hallucinations?
> > Basically, I would like to understand how well this performs when it is in a setting more comparable to the unsupervised baselines mentioned in the paper, since it seems like the current comparison is not truly apples-to-apples, and its unclear how well this method could be used in an actual deployed setting.
> >
> > That being said, I do think this paper is promising and interesting, and if these concerns are addressed, I would be happy to raise my score.

---

> > > ### Author Response · Authors · 2026-04-08
> > >
> > > We thank the reviewer for the detailed and constructive feedback! Below we address the two remaining concerns:
> > >
> > > ### **1. Numbers and Other Entity Types**
> > > We extend our previous rarity analysis with a fine-grained breakdown by NER category (based on [SpaCy](https://spacy.io/) ).
> > > For each entity type, we also split the samples into High/Low-Rarity groups, consistent with the previous rebuttal (W2-Table R5), and evaluate performance separately.
> > >
> > > Results (AUROC, LLaMA-7B, RAGTruth-Avg):
> > >
> > > |NER_Type|%in NER|LBLens_High|LBLens_Low|LBLens_Δ|Fourier_High|Fourier_Low|Fourier_Δ|Example|
> > > |-|-|-|-|-|-|-|-|-|
> > > |PERSON|5.73%|0.8068|0.8184|-0.0116|0.824|0.8164|+0.0076|Steve Austin; Thorson|
> > > |NORP|0.88%|0.8076|0.7981|+0.0094|0.8222|0.7872|+0.0351|Greek; Korean|
> > > |FAC|0.29%|0.8196|0.7887|+0.0309|0.8253|0.7826|+0.0427|Kennedy Center|
> > > |ORG|9.26%|0.8116|0.8149|**-0.0033**|0.8173|0.8217|-0.0043|Medicaid; BLS; FLSA|
> > > |GPE|3.24%|0.8292|0.8383|-0.0091|0.8244|0.8148|+0.0096|Virginia; Alaska|
> > > |LOC|0.28%|0.8173|0.7829|+0.0344|0.8303|0.7834|+0.0469|the Caribbean Sea|
> > > |PRODUCT|2.05%|0.8312|0.7684|+0.0628|0.8798|0.8217|+0.0581|Excel; DoorDash|
> > > |EVENT|0.23%|0.8072|0.7741|+0.0331|0.7749|0.7192|+0.0557|the Civil War; Boston Marathon|
> > > |WORK_OF_ART|0.91%|0.8139|0.7752|+0.0387|0.8059|0.7964|+0.0095|The Anarchy Workout|
> > > |`CARDINAL`|31.85%|0.8138|0.7828|+0.0310|0.8324|0.8061|+0.0263|2; four; 970; half|
> > > |`ORDINAL`|0.42%|0.8259|0.7777|+0.0483|0.7809|0.7637|+0.0172|second;14th|
> > > |`QUANTITY`|5.02%|0.8310|0.7851|+0.0459|0.8259|0.7743|+0.0516|1 cup; 1/4 inch|
> > > |`PERCENT`|4.57%|0.8262|0.7710|+0.0471|0.8209|0.7821|+0.0388|25%|
> > > |`MONEY`|3.62%|0.8220|0.7952|+0.0268|0.8356|0.8063|+0.0293| £23.70; £250 to £500 |
> > > |`DATE`|25.34%|0.8226|0.8070|+0.0156|0.8199|0.8204|**-0.0005**|Friday; July 2003; 1876|
> > > |`TIME`|6.31%|0.8242|0.7822|+0.0421|0.8466|0.7930|+0.0537|15 minutes; 1 pm|
> > >
> > > Across all 16 categories, including numerics (highlighted), Fourier Δ remains consistently near-zero or positive. Overall, these results support the robustness of our method across diverse entity types.
> > >
> > > ### **2. Transfer / Unsupervised Setting**
> > >
> > > To address your concern on this, we carry out both suggested experiments (AUROC, LLaMA-7B).
> > >
> > > **2a. HalluRAG → RAGTruth**
> > >
> > > We use the detector trained on HalluRAG to evaluate RAGTruth without re-fitting.
> > >
> > > |Method|QA|D2T|Summ|Avg|
> > > |-|-|-|-|-|
> > > |LookbackLens|0.5535|0.6024|0.5587|0.5715|
> > > |Laplacian|0.5812|**0.6128**|**0.5635**|0.5858|
> > > |Wavelet|0.5772|0.5833|0.5442|0.5682|
> > > |Fourier|**0.6689**|0.5951|0.5599|**0.6080**|
> > >
> > > Despite distributional shifts between benchmarks, Fourier-High achieves the best average, suggesting frequency-aware features can capture transferable signals.
> > >
> > > **2b. No-Label Instantiation**
> > >
> > > For this experiment, we note that standard clustering methods (e.g., K-Means) struggles in this setting, due to the high dimensionality (2048 for LLaMA-7B) as well as the strong class imbalance, where hallucinated tokens constitute only a small minority (~10%, see Table A1 in paper).
> > >
> > > Instead, this problem is closer to **outlier detection** under an unsupervised learning setting, as hallucinations correspond to sparse outliers that cannot aggregate into coherent clusters. We therefore adopt **LOF** (Local Outlier Factor), a KNN-based purely unsupervised method that identifies outliers by comparing the local density of each point to that of its neighbors.
> > > We apply PCA (d=30) to reduce dimensionality before LOF (n_neighbors=150, contamination=0.05).
> > >
> > >
> > > |Method|RT-QA|RT-D2T|RT-Summ|Avg|
> > > |-|-|-|-|-|
> > > |EigenScore|0.5253|0.5297|0.4989|0.5180|
> > > |ReDeep|0.6364|0.3960|0.5760|0.5361|
> > > |`Laplacian+LOF`|0.6678|0.5587|0.5823|0.6029|
> > > |`Wavelet+LOF`|**0.6684**|**0.5632**|**0.5858**|**0.6058**|
> > > |`Fourier+LOF`|0.6621|0.5483|0.5417|0.5840|
> > >
> > > The results confirm that frequency-aware features alone carry discriminative signals about hallucination, even with unsupervised instantiation.
> > >
> > > **2c. Pseudo-Labels**
> > >
> > > To make fairer comparisons with verification-based methods more apples-to-apples, we use GPT-4o-mini pseudo-labels to fit the detector, matching the LLM backbone used by SelfCheckGPT and RefChecker.
> > >
> > > Note that this approach remains fully unsupervised, as it does not rely on any human-annotated label; pseudo-labels serve only as calibration signals for fair comparison with LLM-as-judge methods. We further include a prompt-based baseline (Prompt) as a direct LLM-as-judge reference.
> > >
> > > |Method|QA|D2T|Summ|Avg|
> > > |-|-|-|-|-|
> > > |SelfcheckGPT|0.6942|0.8026|0.6674|0.7214|
> > > |RefChecker|0.5865|0.6349|0.6080|0.6098|
> > > |Prompt (GPT-4o-mini)|0.6864|0.6932|**0.7155**|0.6984|
> > > |`Laplacian`|**0.8191**|**0.8300**|0.7029|**0.7840**|
> > > |`Wavelet`|0.8021|0.8218|0.6924|0.7721|
> > > |`Fourier`|0.7924|0.8179|0.6842|0.7648|
> > >
> > > The results indicate that our intrinsic features extract complementary signals that LLM-as-judge methods alone cannot capture.
> > >
> > > ---
> > > We thank the reviewer again for helping to strengthen our paper and we will incorporate these improvements into the final version of the paper.

---

### Official Review · Reviewer_7ejR · 2026-03-12

**Soundness:** 2
**Presentation:** 3
**Significance:** 3
**Originality:** 3
**Overall Recommendation:** 4
**Confidence:** 4

**Summary:**

This paper propose a frequency-aware method to detect contextual hallucinations in LLMs. The authors hypothesize that grounded generation exhibits smooth attention distributions, whereas hallucinations are associated with fragmented, rapidly oscillating attention. By regarding attention weights as discrete temporal signals, the authors applies high-pass filters (DFT, DWT, and Laplacian operators) to extract high-frequency components that capture these instabilities. Empirical evaluations demonstrate that the l2​ norm of these high-frequency signals outperforms existing verification-based and aggregate attention-based baselines in detecting hallucinations across multiple datasets (RAGTruth, HalluRAG).

**Compliance With Llm Reviewing Policy:**

Affirmed.

**Final Justification:**

I would like to thank the authors for their exceptionally thorough and constructive rebuttal. The additional mechanism analysis on multi-hop reasoning—specifically the frequency spectrum breakdown demonstrating that valid reasoning operates on low frequencies while hallucinations manifest as high-frequency anomalies—is elegant and perfectly resolves my primary concern. Furthermore, the new empirical results (Qwen tokenizer, LongBenchV2, attention sinks ablation) sufficiently address my questions regarding the method's robustness and generalizability.

While I maintain my position that the theoretical foundations (the i.i.d. assumption and the span-level sliding window) have strict mathematical limitations, I accept these as practical engineering trade-offs. I appreciate the authors' honest acknowledgement of these points and their commitment to adding a dedicated Limitations section.

Given the compelling empirical evidence and the novel mechanistic insights provided during the rebuttal, I am raising my score to a Weak Accept. Please ensure that all newly provided analyses (especially the multi-hop reasoning results, the spectrum energy table, and the Limitations discussion) are fully integrated into the camera-ready version.

**Key Questions For Authors:**

1. How do you conceptually and mathematically reconcile the contradiction in your span-level methodology? Specifically, I would like to understand the rationale behind applying a sliding window average (a low-pass filter) to aggregate features that were exclusively extracted via high-pass filtering to capture transient instabilities.

2. Could you provide further justification or empirical evidence regarding the robustness of the 0.45 DFT cutoff frequency? I am particularly interested in knowing how this threshold and the model performs when applied to fundamentally different tokenizer architectures (e.g., Qwen or Gemma) or in highly extended context windows (e.g., >32k tokens) where attention distributions are naturally diluted.

3. How does the proposed framework distinguish between the erratic attention shifts characteristic of hallucinations and the legitimate, rapid attention jumps required for valid multi-hop reasoning over complex texts?

**Limitations:**

yes

**Strengths And Weaknesses:**

Strengths:
1. I found the fundamental premise of treating attention weights as discrete temporal signals to be highly original and thought-provoking. Shifting the community's focus from static attention allocation (like attention mass) to dynamic spectral analysis provides a genuinely fresh perspective on mechanistic interpretability.

2. I particularly appreciate the rigorous ablation studies, specifically the isolation of context-directed attention versus generated-token attention. The explicit demonstration that high-frequency instability within the source context attention is the primary indicator of hallucination is a valuable contribution to the field.

Weaknesses:
1. While I appreciate the elegant mathematical attempt in Appendix A, I must point out a critical disconnect in the theoretical justification. The proof relies heavily on the assumption that input token embeddings are drawn as independent and identically distributed (i.i.d.) variables (Assumption A.1). This strictly contradicts the inherently autoregressive and highly context-dependent nature of natural language, rendering the theoretical bounds somewhat artificial.

2. There appears to be a glaring methodological contradiction in the span-level evaluation setup. The authors specifically apply high-pass filters to isolate transient, fine-grained high-frequency spikes. However, by immediately employing a sliding window average over 8 tokens for aggregation, they are mathematically applying a low-pass filter to these exact features. This effectively smooths out the very instability the method aims to capture.

3. I have concerns regarding the robustness of the hardcoded 0.45 cutoff frequency used in the Discrete Fourier Transform. As visually evident in Figure A1, the detection performance collapses precipitously as the cutoff approaches the Nyquist limit. This "cliff-edge" sensitivity suggests that the feature might be overfitted to the specific tokenizers and the relatively short context lengths of the evaluated datasets, rather than representing a universal architectural phenomenon.

4. According to Table A1, the average prompt lengths for RAGTruth (QA) and HalluRAG are merely 439.5 and 649.3 tokens, respectively. In modern long-context applications (e.g., 32K or 128K tokens), attention weights are naturally and severely diluted due to the softmax operation over a vast sequence. This dilution fundamentally alters the underlying baseline and shifts the spectral distribution of the attention signal. I suspect that a high-pass filter narrowly tuned to the extreme high-frequency variations of a ~500-token context may fail in long-context scenarios—either by failing to capture any meaningful energy or, conversely, by misclassifying valid, long-range evidence aggregation as hallucination-induced instability.

5. The manuscript lacks a discussion on how the well-documented "Attention Sinks" phenomenon (where massive attention is persistently allocated to initial tokens) impacts the global frequency spectrum calculated during the DFT step.

---

> ### Author Rebuttal · Authors · 2026-03-31
>
> We thank the reviewer for the thoughtful and detailed feedback. We address each concern below.
>
> ## 1. i.i.d. Assumption
>
> We agree the i.i.d. assumption is a simplification. As stated in Line 156, the proof is presented under a "simplified toy setting" to formalize the intuitive link between semantic heterogeneity and attention roughness. We would like to emphasize that
>
> 1) the characterization of any functions’ roughness via high-frequency components is well-studied [1]. Even we do not use strong assumptions, the roughness can be captured by high-frequency, and
>
>  2) The i.i.d assumption is just for a clear close from in discussion,  but our core insight does not rely on it. If replaced by a more realistic formulation of semantic heterogeneity, such as when token relationships are encoded in the geometry of embedding space, including relaxing settings with AR dependencies (such as k-order Markov based on context), the core insight still remains, but leads to more complex, context-dependent results instead of a closed-form bound in K.
> As it is not a theorem paper, we only demonstrate the intuition by an easy setup. We appreciate your insight and we will clarify this in the revision.
>
> [1] Stein & Shakarchi, (2011). Fourier analysis: an introduction.
>
>
> ## 2. Span-Level Aggregation
>
> We agree that the sliding-window aggregation sometimes can be viewed as a “low-pass”-like filtering, but in our case, 1) it is a standard practice in prior works (e.g. Lookback-Lens, ReDeEP). We include it for fair comparison, and
> 2) does not eliminate high-frequency signals, as boundary effects still preserve local variations.
> Therefore, this step should not purely be viewed as a low-pass filter; rather, it estimates the likelihood that a span is overall hallucinated based on the extracted signals. Hence it’s not methodological contradiction.
>
> We further conducted window-size ablations on RAGTruth, where w=8 is optimal, consistent with Lookback-Lens and ReDeEP.
>
> Table R1: w Ablation (AUROC, Fourier, LLaMA-7B)
>
> | w | QA | D2T | Summ | Avg |
> |---|-------|--------|---------|-----|
> | 4 | 0.860 | 0.872 | 0.709 | 0.814 |
> | **8** | **0.873** | **0.887** | **0.764** | **0.841** |
> | 16 | 0.840 | 0.873 | 0.656 | 0.790 |
> | 32 | 0.825 | 0.856 | 0.639 | 0.773 |
>
> ## 3. Robustness:
> **(a) Robust of 0.45 cutoff**– *Nyquist limit and buffer*. In theory, the Nyquist limit (normalized frequency 0.5), rooted in the Nyquist–Shannon sampling theorem, defines the highest representable frequency in discrete signal analysis. Components near this limit correspond to near-maximal oscillation rates that are more likely to reflect noise than structured signal behavior.
> In practice, we need to maintain a buffer zone below the Nyquist limit to exclude such unstable components. Our cutoff of 0.45 provides such a buffer, and Figure A1 confirms its robustness.
>
> **(b) Robust to tokenizers.** We examined the 0.45 cutoff on the suggested Qwen model, which has a different tokenizer.
>
> Table R2: Qwen2.5-7B-Instruct (AUROC)
>
> | Method | QA | D2T | Summ | Avg |
> |--------|-------|--------|---------|-----|
> | Lb-Lens | 0.605 | 0.751 | 0.514 | 0.623 |
> | Lap | 0.606 | 0.760 | 0.583 | 0.650 |
> | Wavelet | **0.615** | 0.759 | 0.575 | 0.650 |
> | Fourier | 0.614 | **0.760** | **0.594** | **0.656** |
>
> **(c) Robust to long-context.** We evaluated on LongBenchV2 (from huggingface) with Qwen-7B.
>
> Table R3: LongBenchV2 (AUROC)
>
> | Method         | Government (20k) | Financial (49k) |
> |----------------|-------------------|-----------------|
> | Lb-Lens  | 0.604          | 0.349          |
> | Lap      | 0.775             | 0.515         |
> | Wavelet   | 0.792          | 0.591          |
> | Fourier   | 0.792          | 0.546          |
>
> While all methods face increased difficulty at longer contexts, high-frequency features remain more informative than Lookback-Lens under significant attention dilution.
>
> ## 4. Attention sinks.
> Attention sinks manifest as a peak at the beginning, present uniformly for both hallucinated and non-hallucinated generations. We preliminarily verified its impact by removing the initial token attention.
>
> Table R4: Origin→Sink-removed (AUROC, LLaMA-7B):
>
> | Method | QA | D2T | Summ |
> |--------|-----|-----|------|
> | Lb-Lens | 0.848→0.765 | 0.844→0.787 | 0.716→0.769 |
> | Fourier | 0.858→0.765 | 0.860→0.801 | 0.743→0.776 |
>
> Results show similar trends across both methods after removal, suggesting attention sinks do not significantly affect the power of our method.
>
>
> ## 5. Distinguishing from Multi-Hop Reasoning
>
> This is indeed a subtle yet important question. Our high-pass filters are designed to capture erratic, rapid attention fluctuation across closely nearby positions, which are characteristic of hallucinated tokens.
> In contrast, in valid multi-hop reasoning, attention shifts across evidence locations in a more structured manner and is therefore not captured by our methods. We will clarify and discuss specifically in the next version.

---

> > ### Author Rebuttal · Reviewer_7ejR · 2026-04-03
> >
> > I would like to thank the authors for their detailed response and the substantial effort put into the rebuttal.
> >
> > I highly commend the additional experimental work provided during the rebuttal period. The evaluations using the Qwen tokenizer, the LongBenchV2 dataset for extreme context lengths, and the attention sink ablation are very valuable. These results effectively alleviate my primary empirical concerns regarding the method's generalizability across different architectures and its robustness in long-context scenarios.
> >
> > Regarding the theoretical justifications, I understand your rationale for using the i.i.d. assumption in a toy setting, and I recognize that applying a sliding window for span-level aggregation is an engineering trade-off necessary to align with existing baselines. However, from a strict signal processing perspective, the span-level aggregation still fundamentally acts as a low-pass filter, which inherently dilutes the pure high-pass motivation. While I accept this as a practical compromise, I strongly request that these theoretical nuances—both the limitations of the i.i.d. relaxation and the mathematical contradiction in the span-level filtering—be explicitly and transparently discussed in the Limitations or Methodology section of the final camera-ready version.
> >
> > Regarding my question on distinguishing the erratic attention shifts of hallucinations from the legitimate attention jumps required for multi-hop reasoning: while your intuitive textual explanation makes sense, this is a highly subtle and critical distinction that warrants empirical validation, not just theoretical conjecture. I would like to see concrete experimental analysis or comparative case studies demonstrating that your high-frequency features do not mistakenly penalize the structured attention shifts inherent in complex, multi-hop reasoning tasks (e.g., using a multi-hop QA dataset like HotpotQA or MuSiQue).
> >
> > If you can provide convincing empirical evidence or quantitative analysis addressing this specific multi-hop reasoning concern, I am open to raising my score.

---

> > > ### Author Response · Authors · 2026-04-08
> > >
> > > Thank you for your thoughtful follow-up and valuable suggestions!
> > > We address both aspects of your comment below.
> > >
> > > ### **1. Addition of Limitations Section**
> > > Regarding the theoretical aspects, we agree that they require clearer discussion. In the revision, we will add a dedicated Limitations section to explicitly discuss (i) the i.i.d. relaxation and (ii) the boundary conditions and potential inconsistencies of span-level filtering.
> > >
> > > ### **2. Empirical Analysis on Multi-Hop Reasoning**
> > > We agree that distinguishing hallucination from legitimate multi-hop reasoning is a subtle and important issue.
> > > To address this concern, we conduct additional experiments on multi-hop QA benchmarks: HotpotQA (7,405 validation examples) and MuSiQue ( 2417 validation examples). The performance of Qwen2.5-7B-Ins is:
> > >
> > > |Data|EM|F1|
> > > |-|-|-|
> > > |HotpotQA|52.49%|67.23%|
> > > |MuSiQue|25.98%|38.67%|
> > >
> > > **Protocol.** We apply Lookback-Lens and our frequency-aware detectors (trained on RT-QA) at token level.
> > > To assess whether valid multi-hop attention shifts are mistakenly flagged as hallucination, we use the EM = 1 subset (exactly correct answers)  as a clean reference, where all answer tokens are guaranteed to be faithful, so any hallucination prediction is a misjudgment.
> > >
> > > We therefore report two complementary quantities:
> > >
> > > - **Total Flagged Tokens**: the number of tokens predicted as hallucinated across the full dataset (to avoid overly conservative detectors).
> > > - **Misjudged on Correct Answers**: the number of misjudged hallucinated tokens on the correct answer .
> > >
> > > **Results.**
> > >
> > > |Method|HotpotQA||MuSiQue||
> > > |-|-|-|-|-|
> > > ||Total Flagged|Misjudged on Correct|Total Flagged|Misjudged on Correct|
> > > |LBLens|2482|3 / 14306 (0.02%)|1912|1 / 2704 (0.04%)|
> > > |Lap|1142|2 / 14306 (0.01%)|749|2 / 2704 (0.07%)|
> > > |Wavelet|1213|3 / 14306 (0.02%)|909|1 / 2704 (0.04%)|
> > > |Fourier|2706|4 / 14306 (0.03%)|1632|1 / 2704 (0.04%)|
> > >
> > > Across both datasets, the misjudgment rate on correct multi-hop reasoning answers is consistently below 0.07%.
> > > This shows that our frequency-aware detectors almost never penalize valid multi-hop reasoning tokens, despite actively flagging hallucination signals.
> > >
> > >
> > > ### **3. A Mechanism Analysis on Spectrum**
> > >
> > > The results above show that multi-hop reasoning is rarely misjudged. To further find the reason behind it, we analyze attention spectra at the token level, providing evidence that hallucination and multi-hop reasoning occupy different frequency bands.
> > >
> > > **Token sets.** We firstly categorize tokens into 3 groups:
> > > - (A) Hallucinated tokens (RT-QA hallucinated answer; N = 3,249),
> > > - (B) Multi-hop tokens (HotpotQA, EM = 1 answer; N = 1,500),
> > > - (C) Grounded tokens (RT-QA, faithful answer as reference; N = 7,882).
> > >
> > > **Attention energy.** For each token we compute the frequency band-restricted ℓ₂ energy `ρ` of its attention (Eq. 6 in paper), on two bands: the top 10% **high band [0.45, 0.50]** that our detector operates on, and a comparison bottom 10% **low band [0, 0.05]**.
> > > To remove cross-dataset scale effects, `ρ` is normalized by the grounded set C in the same band.
> > >
> > > To focus on detector-relevant heads, following Sec. 5.4.2, we aggregate the top-100 heads ranked by absolute classifier coefficients (the most hallucination-indicative heads) using coefficient-weighted averaging.
> > > We report the relative freq-band energy `ρ / ρ(C)` for both high/low -freq, defined as the ratio between mean ρ (A or B) and mean ρ(C), along with effect sizes (Cohen’s d) relative to C. We observe the following results:
> > >
> > > |Token Set|ρ_high / ρ_high(C)|Cohen's *d* (high)|ρ_low / ρ_low(C)|Cohen's *d* (low)|
> > > |-|-|-|-|-|
> > > |A — Hallucinated|**1.019×**|**+0.217 (small ↑)**|1.007×|+0.097 (negligible)|
> > > |B — Multi-hop bridge|0.984×|−0.166 (negligible)|**1.089×**|**+1.273 (large ↑)**|
> > > |C — Grounded (ref)|1.000×|—|1.000×|—|
> > >
> > > >*Note:* All Cohen’s d 95% CIs exclude 0, and all ratio 95% CIs exclude 1 (bootstrap, 2,000 resamples).
> > >
> > > In the high band, hallucinated tokens are elevated above grounded, while multi-hop tokens are statistically indistinguishable from grounded.  In the low band, the pattern is reversed: multi-hop tokens show a *large* increase, while hallucinated tokens are negligible.
> > >
> > > One **possible explanation** is: Valid multi-hop reasoning may involve transitions across distant evidence spans, which manifest as slow, long-range variations in attention, can be captured by low-freq components.
> > > In contrast, hallucination may exhibit rapid, local fluctuations across adjacent positions, captured by high-freq components.
> > > Therefore, the two should diverge from grounded tokens in different frequency bands.
> > >
> > > Since our detector operates on high-freq components, it does not penalize valid multi-hop reasoning, as the two can be separated across frequency bands.
> > >
> > > ---
> > >
> > > We sincerely appreciate the reviewer for these insightful suggestions. These analyses strengthen the paper, and we will include the experiments, analysis, reference, and the Limitations in the final version.
> > >
> > > Many thanks!

---

### Decision · Program_Chairs · 2026-04-30

**Decision:**

Accept (regular)

**Comment:**

Summary:
This paper introduces a frequency-aware perspective attention, a novel method to detect contextual hallucinations in LLMs by analyzing their variation during generation. The key observation is that hallucinated tokens are associated with high-frequency attention energy. The authors therefore developed a lightweight hallucination detector that outperforms strong baselines on RAGTruth and HalluRAG benchmarks.

Justifications:
The proposed idea is novel and interesting. The experiments show a clear improvement in the proposed method over the baselines. The authors provided a detailed and thorough rebuttal that addressed most of the reviewers' questions and concerns. I recommend this paper to be accepted conditioning on adding the additional experimental results and limitations mentioned during the rebuttal period.